# FreeText: Training-Free Text Rendering via Attention Localization and Spectral Glyph Injection

Ruiqiang Zhang [* 1]   Hengyi Wang [* 1]   Chang Liu [1]   Guanjie Wang [1]   Zehua Ma [1]   Weiming Zhang [1]

## Abstract

Large-scale text-to-image (T2I) diffusion models excel at open-domain synthesis but still struggle with precise text rendering, especially for multi-line layouts, dense typography, and long-tailed scripts such as Chinese. Prior solutions typically necessitate costly retraining or impose rigid external layout constraints, often compromising aesthetic quality and flexibility. We propose **Free-Text**, a training-free, plug-and-play framework that improves text rendering by leveraging intrinsic mechanisms of *Diffusion Transformer (DiT)* models. **FreeText** decomposes the problem into *where to write* and *what to write*. For the former, we localize writing regions by extracting token-wise spatial attribution from image-to-text attention, using sink-like tokens as stable spatial anchors and topology-aware refinement to produce high-confidence masks. For the latter, we introduce Spectral-Modulated Glyph Injection (SGMI), which injects a noise-aligned glyph prior with frequency-domain band-pass modulation to strengthen glyph structure and mitigate semantic leakage (rendering the concept instead of the word). Extensive experiments on Qwen-Image, FLUX.1-dev, and SD3 variants across longText-Benchmark, CVTG, and our CLT-Bench show consistent gains in text readability while maintaining semantic alignment and aesthetic quality, with modest inference overhead.

## 1. Introduction

In recent years, large-scale text-to-image (T2I) diffusion models (e.g., Stable Diffusion (Esser et al., 2024), FLUX

[1] Anhui Province Key Laboratory of Digital Security, University of Science and Technology of China. Correspondence to: Zehua Ma < mzh045@ustc.edu.cn>, Weiming Zhang <zhangwm@ustc.edu.cn>.

*Proceedings of the 43rd International Conference on Machine Learning*, Seoul, South Korea. PMLR 306, 2026. Copyright 2026 by the author(s).

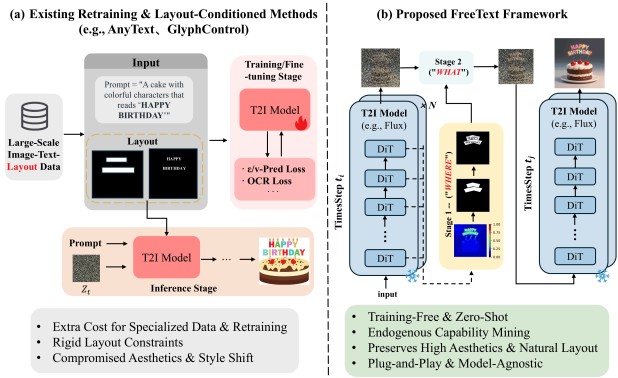

*Figure 1.* System overview. (a) Prior text-rendering methods typically require retraining and/or rigid layout conditions. (b) FreeText decomposes text rendering into *WHERE* and *WHAT*: it localizes text regions via attention maps, then injects a glyph-structure prior in a model-compatible way, enabling training-free enhancement while preserving the base model's aesthetics.

(Labs et al., 2025), and Qwen-Image (Wu et al., 2025)) have achieved strong open-domain image synthesis quality. However, precise text rendering remains challenging, with typos, missing strokes, distortions, and "semantic leakage" (rendering the concept instead of the word), especially in multi-line, text-dense, multilingual, and semantically complex scenes. The issue is particularly severe for logographic scripts, such as Chinese: the character distribution is highly long-tailed, with many rare characters and low-frequency compositions underrepresented during training. Meanwhile, numerous characters are visually similar, featuring complex internal radicals and stroke patterns (Chen et al., 2021). As a result, models often fail to learn reliable glyph priors from limited coverage and are prone to fine-grained confusion, making the rendered text frequently unusable even after repeated sampling. From both application and research perspectives, text rendering is not a cosmetic add-on but a key stress test for fine-grained controllability, complex scene planning, and cross-modal alignment in T2I models. Text is a highly structured visual object whose strokes, glyph shapes, and arrangements impose strict local geometry and global layout constraints. Moreover, humans are susceptible to textual errors. In real-world scenarios such as posters and UI design, text often serves as a crucial identifier, and typos

or malformed glyphs can severely degrade usability. Therefore, better text rendering is essential for practical usability, where minor typos can invalidate an otherwise good image.

Most existing approaches to improve text rendering rely on two strategies: additional training or fine-tuning (retraining-based) and explicit layout or position conditions (layout-conditioned). Methods such as TextDiffuser (Chen et al., 2023) and AnyText (Tuo et al., 2023) train layout predictors or control branches with box/mask/glyph supervision, improving controllability and OCR accuracy. However, these strategies entail significant computation and data costs. Furthermore, extensive fine-tuning often induces a distribution shift and a visual style that deviate from the base model. At inference time, they further inject bounding boxes, masks, or glyphs as hard conditions, mechanically fixing text regions to preset positions. Such external constraints can suppress the model's intrinsic scene-planning behavior, making it difficult to balance diversity and naturalness under complex backgrounds or ambiguous/conflicting prompts.

Meanwhile, large fully pre-trained and extensively post-trained models (e.g., FLUX and Qwen-Image) already exhibit strong aesthetic quality; imposing rigid layout constraints on them is not only difficult to fine-tune but can also noticeably damage their aesthetic quality. Conversely, text-specialized models such as AnyText, which are trained primarily for text rendering, typically cannot replicate the full pre-training and post-training pipelines of large foundation models, and thus often trade off rendering accuracy against aesthetics, with the latter lagging in complex open-domain scenes. To date, there progress remains limited progress on simultaneously achieving high rendering accuracy and strong aesthetics by leveraging only the base model's internal mechanisms, without modifying architectures or parameters.

Motivated by these limitations, we propose a new perspective: instead of paying the high cost of retraining to teach models a generic *how to write*, we decompose text rendering into two more fundamental subproblems that the base model already has the potential to support: *where to write* and *what to write*. Based on this view, we introduce FreeText, a training-free, plug-and-play enhancement framework. The design is driven by a simple fact: for both humans and models, understanding text is easier than generating it with correct glyph structure and topology. FreeText exploits easily accessible visual priors of text and the model's internal structure to address these two subproblems.

1. *WHERE to write.* T2I models are not necessarily lacking layout planning; rather, we have not effectively utilize their latent spatial intentions. In fact, during generation, diffusion models with DiT-style architectures implicitly encode spatial correspondence for different text tokens in image-to-text cross-attention (Peebles & Xie, 2023). Attention maps across timesteps and network depths jointly describe the model's endogenous layout. Based on this, we propose an unsupervised localization strategy: instead of relying on fragile external OCR or Vision-Language Model(VLM) detectors for post-hoc detection, FreeText selects the most stable attention layers as spatial anchors and precisely locks the writing regions for target text tokens under zero layout annotations (zero-layout supervision).

2. *WHAT to write.* As illustrated in Appendix Fig.9, models may render the token "Car" as a visual concept rather than literal word form. We attribute this to the coupling between semantic concepts (high-level meaning) and glyph structures (visual form) in the embedding space. Early in generation, strong concrete semantic priors can dominate and suppress glyph information, causing semantic leakage—i.e., concepts overwhelm strokes and lead to "text becoming images". To enforce the local priority "Glyph > Semantics", we propose Spectral-Modulated Glyph Injection (SGMI). Instead of naively mixing latents, SGMI applies band-pass modulation in the frequency domain to enhance mid-to-high frequency components that carry glyph structures, while suppressing the propagation of background and irrelevant noise, thereby guiding accurate glyph synthesis.

In summary, our contributions are:

- A training-free, model-agnostic text rendering enhancement framework. FreeText operates as an inference-time plug-in, seamlessly integrating into T2I models without modification and substantially elevating text rendering performance in bilingual (Chinese/English) and challenging rendering scenarios.

- An unsupervised text-region localization method based on attention. We leverage DiT-style image-to-text attention signals and an Attention Sink-like stability cue to achieve generic and high-precision text-region locking without any supervision.

- A frequency-domain glyph prior injection mechanism. SGMI utilizes band-pass spectral modulation to selectively amplify structure-carrying glyph frequencies while suppressing semantic-background leakage, improving rendering fidelity.

- A Chinese long-tail text rendering benchmark. We introduce CLT-Bench, a graded evaluation benchmark targeting long-tail Chinese characters (rare and structurally complex) to systematically assess performance degradation from common to rare, and from simple to complex settings.

## 2. Related Work

### 2.1. T2I diffusion foundation models

Recent large-scale T2I diffusion models have steadily improved resolution, semantic alignment, and text rendering (Wu et al., 2025; Seedream et al., 2025; Esser et al., 2024; Labs et al., 2025). Representative systems such as Stable Diffusion 3, Qwen-Image, and FLUX.1 attribute these gains to stronger MMDiT/DiT backbones, flow/rectified-flow objectives, and large dedicated data pipelines, resulting in better overall visual quality and typography. However, such improvements typically require costly pre-training and post-training, and are tightly coupled to specific architectures and data recipes, making text-rendering capability hard to transfer across base models at low cost. In contrast, FreeText keeps the base model unchanged and performs inference-time control by leveraging attention and latent-space structure, enabling cross-model, fine-grained text rendering enhancement.

### 2.2. Retraining and layout-dependent text rendering

Most prior text-rendering methods follow a retraining-based, layout-dependent paradigm. TextDiffuser-style (Chen et al., 2023) approaches learn layout prediction modules on large OCR-annotated corpora, requiring explicit layout templates or segmentation priors at generation time. Methods such as AnyText (Tuo et al., 2023), GlyphDraw (Ma et al., 2023), GlyphControl (Yang et al., 2023), and UniGlyph (Yang et al., 2023) introduce ControlNet-style or dedicated conditional branches on top of Stable Diffusion/DiT, retraining with extra inputs (e.g., glyph images, text masks, or segmentation maps) to improve OCR accuracy and font controllability. While effective, these methods rely on additional annotations and control branches, tightly binding generation to external layout/visual conditions, limiting prompt freedom and image diversity, and underutilizing the base model's endogenous scene planning.

### 2.3. Attention sinks

Attention sinks—where semantically weak tokens absorb disproportionate attention to buffer global context—are well-studied in LLMs (Tigges et al., 2023; Razzhigaev et al., 2025; Chauhan et al., 2025; Zhang et al., 2025). Related analyses in multimodal models use attention patterns to study cross-modal alignment and hallucination (Kang et al., 2025). Yet, attention sinks have rarely been systematically used for text-region localization in T2I diffusion models. FreeText empirically finds that sink-like tokens in DiT-based T2I models produce stable boundary cues across timesteps and layers; we treat them as spatial anchors to extract text regions from attention without supervision, enabling reliable glyph injection.

## 3. Method

FreeText aims to enhance text rendering in complex scenes without modifying the architecture or parameters of the base transformer-based T2I diffusion model. Given a target text span $s$ and its glyph reference image, FreeText proceeds in two stages, as shown in Fig. 2.

1. **Attention-guided text-region localization** (Sec. 3.1): we extract image-to-text cross-attention from DiT/M-MDiT blocks during sampling, aggregate and select informative timestep–layer pairs, and apply topology-aware post-processing to obtain a high-confidence writing mask $\mathbf{R}_s$ in latent space.

2. **Spectral-Modulated Glyph Injection** (Sec. 3.2): we encode the glyph reference into latent space, align it to the current noise level, construct a Log-Gabor based Spectral-Modulated Glyph Injection (SGMI) prior, and inject it into $\mathbf{R}_s$ within a short time window using cosine annealing, strengthening glyph structure and suppressing semantic leakage (e.g., rendering the concept instead of the word).

### 3.1. Attention-guided text-region localization

To answer "where to write", we localize the writing region directly from attention (Tang et al., 2022), without external layout predictors, OCR, or VLM detectors. We read out token-wise spatial attribution from attention maps, then perform timestep–layer selection and topology-aware refinement to produce a high-confidence region mask.

#### 3.1.1. ATTENTION EXTRACTION

Let $\mathbf{A}^{(t,l)}$ denote the head-averaged I2T attention at timestep $t$ and the $l$-th DiT/MMDiT block:

$$\mathbf{A}^{(t,l)} \in \mathbb{R}^{H \times W \times N_{\text{text}}}, \tag{1}$$

where $N_{\text{text}}$ is the number of text tokens. For a target span $s$, we first locate its token subsequence $\mathcal{T}_s$, and augment it with a few sink-like special tokens that exhibit stable high responses across layers/heads. We call the union the anchor token set $\tilde{\mathcal{T}}_s$.

We then average attention over $\tilde{\mathcal{T}}_s$ to obtain an initial localization map:

$$\mathbf{M}^{(t,l)}(x,y) = \frac{1}{|\tilde{\mathcal{T}}_s|} \sum_{k \in \tilde{\mathcal{T}}_s} \mathbf{A}^{(t,l)}_{x,y,k}, \tag{2}$$

and linearly normalize $\mathbf{M}^{(t,l)}$ to $[0,1]$. For clarity, we omit the subscript $s$ in what follows.

#### 3.1.2. TIMESTEP-LAYER SELECTION

As shown in Fig. 3, naively aggregating attention across all timesteps and blocks introduces substantial noise: early

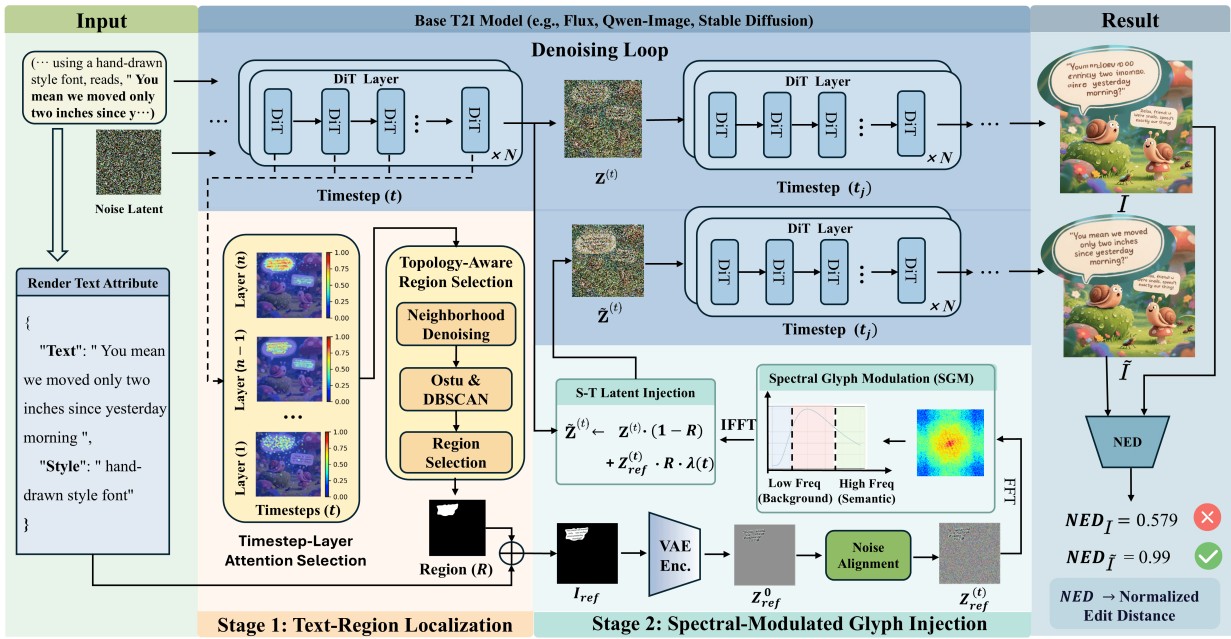

*Figure 2.* Overview of FreeText.

steps are coarse and reflect global planning; mid steps are most informative for writing placement; late steps become diffuse due to global refinement (Chefer et al., 2023; Darcet et al., 2024). In addition, shallow blocks emphasize local geometry while deeper blocks integrate global semantics. We therefore select informative timestep-layer pairs before aggregation.

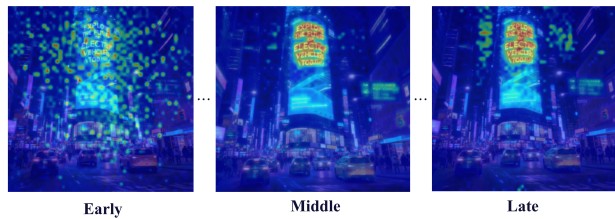

*Figure 3.* Typical I2T attention patterns across timesteps: early steps are coarse, mid steps concentrate on target regions, and late steps become diffuse.

Given candidate sets $\mathcal{T}_{\text{cand}}$ and $\mathcal{L}_{\text{cand}}$, we score each pair $(t, l)$ using a *soft IoU* between $\mathbf{M}^{(t,l)}$ and a reference mask $\mathbf{Y} \in [0, 1]^{H \times W}$:

$$\text{IoU}(t, l) = \frac{\langle \mathbf{M}^{(t,l)}, \mathbf{Y} \rangle}{\|\mathbf{M}^{(t,l)}\|_1 + \|\mathbf{Y}\|_1 - \langle \mathbf{M}^{(t,l)}, \mathbf{Y} \rangle}. \quad (3)$$

We select the top-$K$ pairs to form $\mathcal{S}$ and aggregate:

$$\mathbf{M}(x, y) = \frac{1}{|\mathcal{S}|} \sum_{(t,l) \in \mathcal{S}} \mathbf{M}^{(t,l)}(x, y). \quad (4)$$

### 3.1.3. TOPOLOGY-AWARE REGION SELECTION

The aggregated map $\mathbf{M}$ may still contain isolated peaks and fragmented clusters. We apply a lightweight post-processing pipeline to produce the final writing mask.

We first perform local neighborhood aggregation on $\mathbf{M}$ to suppress small outliers and promote connected responses. Next, we binarize $\mathbf{M}$ into $\mathbf{B} \in \{0, 1\}^{H \times W}$ using an adaptive threshold selected by maximizing inter-class variance (Otsu et al., 1975). We then run DBSCAN (Ester et al., 1996) on foreground pixels to obtain candidate connected regions $\{\mathcal{C}_i\}$ while discarding sparse noise.

Each region $\mathcal{C}_i$ is scored on the original $\mathbf{M}$:

$$q_i = \frac{|\{(x, y) \in \mathcal{C}_i \mid \mathbf{M}(x, y) > \tau\}|}{|\mathcal{C}_i|}, \quad (5)$$

where $\tau$ is set as a high quantile of $\mathbf{M}$ within the union of candidate regions. We select the best region and resize it to latent resolution to obtain the binary writing mask:

$$\mathbf{R} \in \{0, 1\}^{H_{\text{lat}} \times W_{\text{lat}}}. \quad (6)$$

In Sec. 3.2, $\mathbf{R}$ is broadcast across channels for local latent injection.

### 3.2. Spectral-Modulated Glyph Injection

To answer "what to write", we enhance glyph structure while suppressing semantic leakage. We encode a glyph reference into latent space, align it to the current noise level, apply Log-Gabor based SGMI to emphasize structure-carrying

frequencies, and inject the resulting prior into $\mathbf{R}$ within a short time window.

### 3.2.1. NOISE-ALIGNED LATENT PROJECTION

We rasterize the target text $s$ into a glyph reference image $\mathbf{I}_{\text{glyph}}$ placed in region $\mathbf{R}$, and encode it with the same VAE as the base model:

$$\mathbf{z}_{\text{ref}} = E_{\text{VAE}}(\mathbf{I}_{\text{glyph}}) \in \mathbb{R}^{C \times H_{\text{lat}} \times W_{\text{lat}}}. \tag{7}$$

At timestep $t$ with noise schedule $(\alpha_t, \sigma_t)$, we match the noise level via forward diffusion:

$$\mathbf{z}_{\text{ref}}^{(t)} = \alpha_t \mathbf{z}_{\text{ref}} + \sigma_t \boldsymbol{\epsilon}, \quad \boldsymbol{\epsilon} \sim \mathcal{N}(0, \mathbf{I}). \tag{8}$$

### 3.2.2. LOG-GABOR SPECTRAL MODULATION

On $\mathbf{z}_{\text{ref}}^{(t)}$, we apply a Log-Gabor filter (Field, 1987) to strengthen mid-to-high frequencies that carry glyph structure while suppressing low-frequency background and ultra-high-frequency noise. Let $G(\rho, \theta)$ be the Log-Gabor kernel in the 2D frequency domain. For each channel $c$:

$$\widehat{\mathbf{z}}_{\text{ref},c}^{(t)} = \mathcal{F}\left(\mathbf{z}_{\text{ref},c}^{(t)}\right), \tag{9}$$

$$\widehat{\mathbf{z}}_{\text{sgmi},c}^{(t)}(\rho, \theta) = G(\rho, \theta) \cdot \widehat{\mathbf{z}}_{\text{ref},c}^{(t)}(\rho, \theta), \tag{10}$$

$$\mathbf{z}_{\text{sgmi},c}^{(t)} = \mathcal{F}^{-1}\left(\widehat{\mathbf{z}}_{\text{sgmi},c}^{(t)}\right), \tag{11}$$

where $\mathcal{F}$ and $\mathcal{F}^{-1}$ are 2D FFT and inverse FFT. The resulting $\mathbf{z}_{\text{sgmi}}^{(t)}$ is the SGMI-enhanced reference latent at timestep $t$.

### 3.2.3. ANNEALED SPATIOTEMPORAL INJECTION

Let the sampling trajectory evolve from timestep $T$ to 0. We inject glyph priors only in a mid-early window:

$$t_{\text{start}} = 0.8T, \quad t_{\text{end}} = 0.6T, \tag{12}$$

to avoid disrupting early global planning or late-stage fine-detail refinement. For $t$ within the injection window, we define a cosine-annealed weight:

$$\lambda(t) = \frac{1}{2}\left(1 + \cos\left(\pi \cdot \frac{t - t_{\text{start}}}{t_{\text{end}} - t_{\text{start}}}\right)\right), \tag{13}$$

and update the denoising latent $\mathbf{z}^{(t)}$ by masked replacement (Avrahami et al., 2023):

$$\tilde{\mathbf{z}}^{(t)} = \left(\mathbf{I} - \lambda(t)\mathbf{R}\right) \odot \mathbf{z}^{(t)} + \lambda(t)\mathbf{R} \odot \mathbf{z}_{\text{sgmi}}^{(t)}. \tag{14}$$

For $t \notin [t_{\text{start}}, t_{\text{end}}]$, we keep $\mathbf{z}^{(t)}$ unchanged.

### 3.3. CLT-Bench: Chinese long-tail text rendering

Chinese text rendering is challenging due to a long-tailed character distribution and high intra-class visual similarity. Existing benchmarks over-emphasize common characters and/or English, obscuring degradation from frequent/simple to rare/complex cases (Zhao et al., 2025; Fang et al., 2025; Du et al., 2025). We introduce CLT-Bench to stress-test T2I text rendering under rare-character and complex-layout settings.

We assign each prompt a complexity score combining character difficulty and layout difficulty. For a character $c$, we normalize stroke count $s(c)$ and frequency rank $r(c)$:

$$S(c) = \frac{s(c) - s_{\min}}{s_{\max} - s_{\min}}, \quad R(c) = \frac{r(c) - r_{\min}}{r_{\max} - r_{\min}}, \tag{15}$$

and define character difficulty

$$D(c) = \frac{w_s S(c) + w_f R(c)}{w_s + w_f} \in [0, 1]. \tag{16}$$

Given text segments $\{\text{txt}_i\}_{i=1}^{N_{\text{seg}}}$ with characters $\{c_j\}_{j=1}^{N_{\text{chars}}}$, we compute

$$C_{\text{char}} = \frac{1}{N_{\text{chars}}} \sum_j D(c_j),$$

$$C_{\text{len}} = \min\left(\frac{N_{\text{chars}}}{N_{\max}}, 1\right), \tag{17}$$

$$C_{\text{seg}} = \min\left(\frac{N_{\text{seg}} - 1}{M_{\max} - 1}, 1\right),$$

where $N_{\max}$ is a preset upper bound on the total number of characters to render in a prompt, and $M_{\max}$ is a preset upper bound on the number of text segments (regions) to render. The prompt score is then

$$Score = \frac{w_{\text{char}} C_{\text{char}} + w_{\text{len}} C_{\text{len}} + w_{\text{seg}} C_{\text{seg}}}{w_{\text{char}} + w_{\text{len}} + w_{\text{seg}}} \in [0, 1]. \tag{18}$$

We stratify prompts by *Score* to form subsets spanning common/simple to rare/complex characters with challenging multi-segment layouts.

## 4. Experiments

### 4.1. Experimental setup

#### 4.1.1. BASE MODELS

We evaluate FreeText on four representative T2I foundation models: (i) Qwen-Image (Chinese/English prompts), (ii) FLUX.1-dev (English only), (iii) Stable Diffusion 3.5 Large (SD3.5-L; English only), and (iv) Stable Diffusion 3 Medium (SD3-M; English only). All experiments compare *Base* vs. *Base + FreeText*. FreeText is used as an inference-time plug-in: it does not modify model parameters, architectures, or introduce learnable branches.

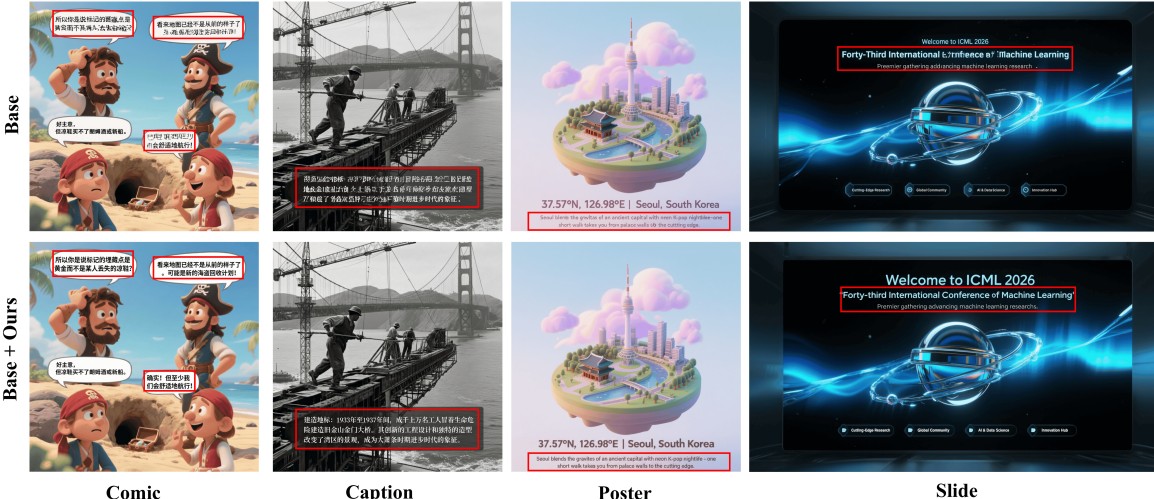

*Figure 4.* Baseline comparison across four text-rendering scenarios (comic, caption, poster, slide). Top: Base; bottom: Base+FreeText. Red boxes highlight the target text regions, where FreeText reduces typos/malformed glyphs and improves readability.

### 4.1.2. BENCHMARKS AND PROTOCOL

We use three benchmarks covering long text, multi-region rendering, and long-tail Chinese: (1) **longText-Bench** with longText-en/zh, focusing on long prompts and paragraph-level, multi-line text (Geng et al., 2025); (2) **CVTG**, with 2/3/4/5 text regions (2–5 segments) and typically short prompts (Du et al., 2025); (3) **CLT-Bench** (Sec. 3.3), targeting rare and structurally complex Chinese characters.

**Benchmark–model compatibility.** Qwen-Image and FLUX.1-dev are evaluated on longText-Benchmark and CVTG. SD3.5-L and SD3-M are evaluated on CVTG only, since long prompts can be truncated by their text encoders. CLT-Bench is evaluated on Qwen-Image only.

**Inference settings.** Unless noted, Base and Base + FreeText use identical resolution, sampling steps, and sampler hyperparameters. FreeText uses the default annealed injection window (Sec. 3.2); in this section we refer to the injection module as **SGMI**.

**Metrics.** We measure both text readability and overall image quality (higher is better unless noted): **NED** (Normalized Edit Distance, via a fixed OCR engine (Cui et al., 2025)), **CLIPScore** (text–image alignment), **AestheticScore** (LAION aesthetic predictor), and **VQA Score** (VLM-based usability/clarity QA; templates in the appendix). For localization analysis, we report **IoU** between predicted and reference text regions.

### 4.2. Effectiveness of FreeText

#### 4.2.1. QWEN-IMAGE AND FLUX.1-DEV

Table 2 reports results on longText-Benchmark and CVTG. FreeText consistently improves NED and VQA Score (Fang

*Table 1.* End-to-end results on CVTG for SD3 models.

| Model | Setting | NED↑ | CLIP↑ | Aes↑ | VQA↑ |
|---|---|---|---|---|---|
| SD3.5-L | Base | 0.848 | **0.879** | 5.634 | 3.849 |
| SD3.5-L | Base + FreeText | **0.864** | 0.871 | 5.608 | **4.595** |
| SD3-M | Base | 0.616 | 0.851 | 5.906 | 2.903 |
| SD3-M | Base + FreeText | **0.669** | **0.852** | **5.917** | **3.674** |

et al., 2025), indicating higher text readability, while CLIP-Score and AestheticScore remain largely stable, suggesting limited impact on semantic alignment and aesthetics.

### 4.2.2. SD3-M AND SD3.5-L

Since SD3 variants are sensitive to long prompts, we evaluate them on CVTG only (Table 1). FreeText improves NED and VQA Score for both models, while CLIPScore and AestheticScore remain comparable, indicating the local SGMI injection does not introduce notable semantic leakage or quality degradation.

### 4.2.3. CLT-BENCH

On CLT-Bench (Qwen-Image only), FreeText improves NED but with smaller gains (Table 3). This suggests **FreeText** is most effective when pretraining has endowed the base model with a basic prior for the target characters, while the remaining bottleneck lies in high-fidelity glyph generation. **FreeText** primarily strengthens glyph structure, rather than enabling genuinely unseen characters from scratch.

### 4.2.4. BENEFIT PROPAGATION UNDER FULL ATTENTION

We observe cross-region benefit propagation: correcting one text region with FreeText can improve other regions that are

*Table 2.* End-to-end results on longText-Benchmark and CVTG.

| Model | Setting | Subset | NED↑ | CLIP↑ | Aes↑ | VQA↑ |
|-------|---------|--------|------|-------|------|------|
| Qwen-Image | Base | longText-en | 0.625 | 0.858 | 4.912 | 2.650 |
| Qwen-Image | Base + FreeText | longText-en | **0.713** | **0.864** | **5.013** | **4.177** |
| FLUX.1-dev | Base | longText-en | 0.598 | 0.863 | **5.365** | 2.563 |
| FLUX.1-dev | Base + FreeText | longText-en | **0.690** | **0.868** | 5.342 | **4.211** |
| Qwen-Image | Base | longText-zh | 0.639 | 0.474 | 4.607 | 3.657 |
| Qwen-Image | Base + FreeText | longText-zh | **0.694** | **0.537** | **4.749** | **4.211** |
| Qwen-Image | Base | CVTG | 0.574 | 0.781 | 4.386 | 2.756 |
| Qwen-Image | Base + FreeText | CVTG | **0.619** | **0.794** | **4.391** | **3.469** |
| FLUX.1-dev | Base | CVTG | 0.712 | 0.836 | 5.910 | 4.050 |
| FLUX.1-dev | Base + FreeText | CVTG | **0.722** | **0.839** | **5.936** | **4.952** |

not explicitly processed, reflected by higher global metrics (e.g., VQA Score). We attribute this to global self-attention in DiT/MMDiT: patch tokens mix information globally at each denoising step, so severe errors in one region can perturb updates elsewhere; once a key error is corrected, this interference is reduced.

*Table 3.* End-to-end NED on CLT-Bench.

| Model | Setting | NED↑ |
|-------|---------|------|
| Qwen-Image | Base | 0.458 |
| Qwen-Image | Base + FreeText | **0.488** |

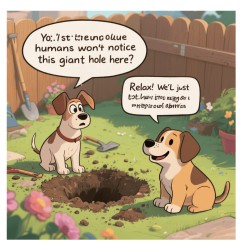
(a) Base

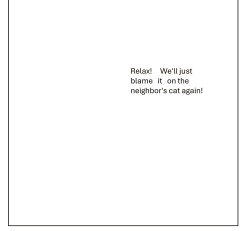
(b) Inject prior (Text-2 only )

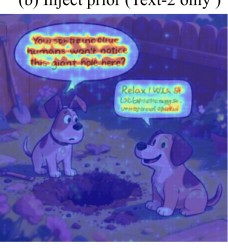
(c) Base + ours

(d) Attn while generating Text-1

*Figure 5.* Cross-region benefit propagation and attention evidence (example with two text lines; refining only one line can improve the other).

### 4.2.5. TOKEN CHOICE

We compare three token sets for each target span: **Entity-only** (tokens of the target string), **Sink-only** (sink-like special tokens), and **Entity + Sink**. As shown in Table 5 and

Fig. 6, Sink-only is more temporally stable but has a lower ceiling, while Entity + Sink achieves the best IoU by combining explicit semantic attribution with stable sink responses, yielding more reliable masks for SGMI.

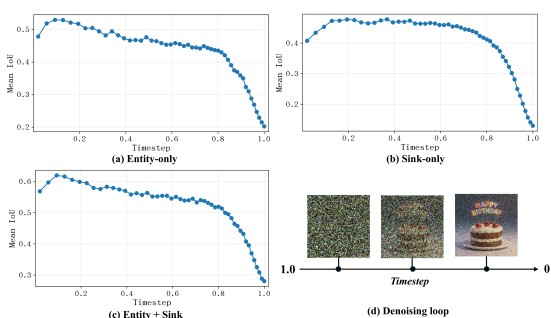

*Figure 6.* IoU vs. timestep for different token sets.

### 4.3. Localization strategy

We next analyze the reliability of the *WHERE* stage.

#### 4.3.1. COMPARISON WITH VLM-BASED LOCALIZATION

Table 4 compares our endogenous localization against several closed-source VLM baselines. In practice, multi-line text, cluttered backgrounds, and malformed glyphs can break "recognize-then-localize" pipelines; recognition failure often cascades into localization failure. By reading I2T attention directly, FreeText avoids this chain and provides a more stable signal.

*Table 4.* Localization IoU comparison.

| Method | IoU↑ |
|--------|------|
| doubao-seed-1-6-251015 | 0.325 |
| gemini-2.5-flash-lite | 0.139 |
| gpt-5.1 | 0.159 |
| qwen3-vl-plus-2025-09-23 | 0.195 |
| FreeText (ours) | **0.561** |

*Table 5.* Localization IoU for different token sets.

| Setting | IoU↑ |
|---|---|
| Entity-only | 0.495 |
| Sink-only | 0.479 |
| Entity + Sink | **0.561** |

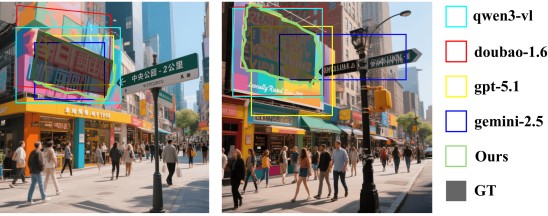

- qwen3-vl
- doubao-1.6
- gpt-5.1
- gemini-2.5
- Ours
- GT

*Figure 7.* Typical VLM localization failures under multi-line text and degraded glyphs, compared with endogenous localization.

### 4.4. Ablation study

We ablate the *WHAT* stage(SGMI). We compare three variants: **B** (Base), **+F** (Base+FreeText), and **+F− SGMI** (Base+FreeText without SGMI, i.e., removing the spectral band-pass modulation while keeping the rest unchanged). As shown in Table 6, removing SGMI reduces NED and VQA Score, while CLIP and Aes remain largely unchanged, indicating SGMI primarily contributes to text readability improvements. As further illustrated in Fig. 8, injecting only low-frequency components loses stroke-level structure, while injecting only high-frequency components (where semantics dominates) can trigger concept-texture intrusion. In contrast, SGMI's band-pass design provides an injection signal that is most effective for glyph structure while being most conservative against semantic leakage. This indicates that the key of frequency-domain modulation is not *injecting more information*, but *injecting the right spectral band*.

*Table 6.* Ablation on SGMI.

| Model | Settings | NED↑ | CLIP↑ | Aes↑ | VQA↑ |
|---|---|---|---|---|---|
| Qwen-Image | B | 0.625 | 0.858 | 4.912 | 2.650 |
|  | +F− SGMI | 0.686 | 0.860 | **5.027** | 3.724 |
|  | +F | **0.713** | **0.864** | 5.013 | **4.177** |
| FLUX.1-dev | B | 0.598 | 0.863 | **5.365** | 2.563 |
|  | +F− SGMI | 0.671 | 0.865 | 5.361 | 3.816 |
|  | +F | **0.690** | **0.868** | 5.342 | **4.211** |

### 4.5. Efficiency

We measure inference overhead on an NVIDIA A6000 with bfloat16, resolution $928 \times 928$, and 50 sampling steps. Table 7 shows that FreeText adds moderate overhead (primarily from Stage-1 localization, which accumulates and selects I2T attention before injection), increasing latency by roughly 12%–18% with $< 1$GB peak-memory overhead.

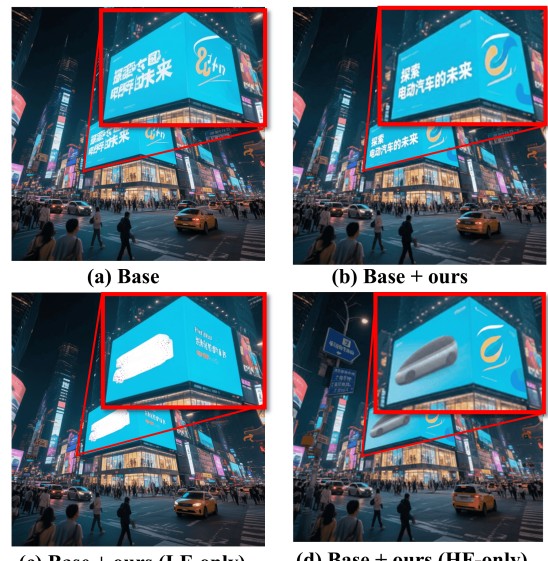

| (a) Base | (b) Base + ours |
|---|---|
| (c) Base + ours (LF-only) | (d) Base + ours (HF-only) |

*Figure 8.* Qualitative ablation illustrating semantic leakage and stroke degradation under different spectral settings.

*Table 7.* Inference efficiency.

| Model | Setting | Time (s)↓ | Mem (GB)↓ |
|---|---|---|---|
| Qwen-Image | Base | **37.64** | **53.76** |
| Qwen-Image | Base + FreeText | 42.33 | 54.35 |
| FLUX.1-dev | Base | **41.56** | **31.44** |
| FLUX.1-dev | Base + FreeText | 47.17 | 32.17 |
| SD3.5-L | Base | **35.03** | **26.11** |
| SD3.5-L | Base + FreeText | 41.17 | 26.91 |
| SD3-M | Base | **9.85** | **14.53** |
| SD3-M | Base + FreeText | 11.47 | 14.97 |

## 5. Conclusion

We presented FreeText, a training-free and plug-and-play framework for improving text rendering in T2I diffusion models without changing model weights or architectures. By decomposing text rendering into *where to write* and *what to write*, FreeText (i) localizes writing regions from endogenous attention via sink-anchored, topology-aware selection, and (ii) enhances glyph fidelity through SGMI, a noise-aligned frequency-domain injection that strengthens structure-carrying components and mitigates semantic leakage. Across multiple foundation models and benchmarks, FreeText consistently improves readability metrics while maintaining CLIPScore and AestheticScore, and incurs only moderate runtime and memory overhead. Future research will focus on validating the universality of our approach by adapting it to diverse emerging foundation models.

## Acknowledgements

This work was supported in part by the Natural Science Foundation of China under Grant 62402469, U2336206, U2436601, 62121002.

## Impact Statement

This paper presents work whose goal is to advance the field of Machine Learning. There are many potential societal consequences of our work, none which we feel must be specifically highlighted here.

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

## A. Semantic leakage: Rendering Concepts Instead of Words

As shown in Fig. 9 (a), the two target lines, "Copper Sulfate Solution" and "0.5M Concentration," are not rendered verbatim on the label. Instead, the model interprets the input as the concept of a copper sulfate solution and replaces the requested strings with a scene-plausible depiction, causing a mismatch in both the exact wording and the intended layout (two lines and the specific concentration).

As shown in Fig. 9 (b), the Chinese phrase "前方弯路，减速慢行" is also not reproduced character-by-character. The model directly maps the semantics to a canonical traffic-sign symbol associated with "slow down," substituting a symbolic concept for the required textual content. Together, these examples illustrate a typical case of semantic leakage: the output remains semantically related to the prompt, yet fails to faithfully render the specified characters.

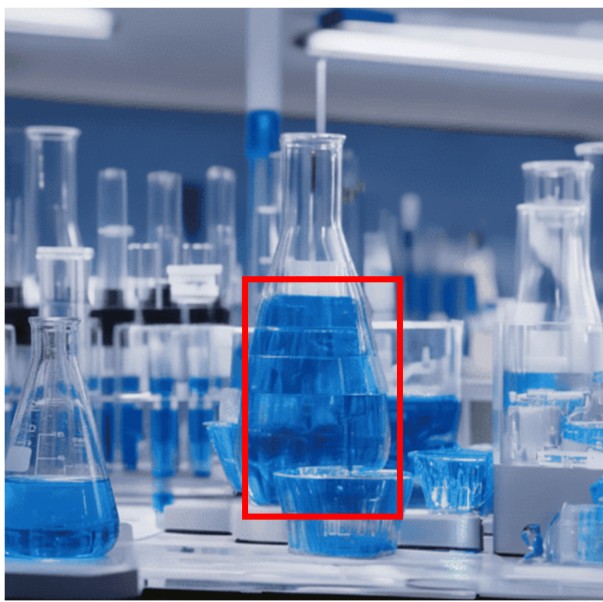 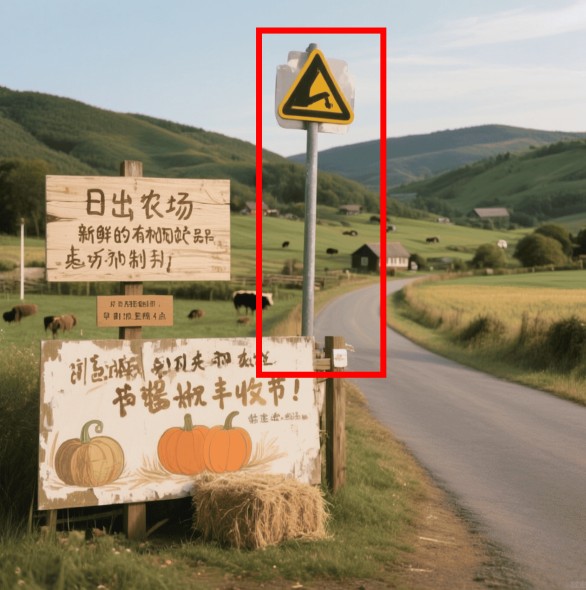

*Figure 9.* Examples of semantic leakage in text rendering, where the model depicts the underlying concept instead of faithfully reproducing the target strings. (a) The requested label text is conceptualized, drifting from the exact wording and the intended two-line layout. (b) The Chinese warning phrase is replaced by a canonical "slow down" traffic-sign symbol rather than the specified characters.

---

**Prompts used in Fig. 9**

**(a) Prompt.**

. . . At the center of the scene prominently stands a transparent glass flask containing a liquid, clearly labeled in concise, bold lettering as "**Copper Sulfate Solution**", complemented neatly just below by smaller, precisely printed text reading "**0.5M Concentration**". . . .

**(b) Prompt.**

一条宁静的乡村道路通向一家小型家族农场的入口，这里设有多种标识。...道路拐弯处的金属交通标识牌以明黄色和黑色警示：前方弯路，减速慢行。这些标识清晰易读，周围环绕着连绵起伏的绿色山丘、放牧的牲畜和零星分布的农舍，构成了一幅宁静宜人的田园景象。

## A.1. Additional Qualitative Examples with English Prompts

## A.2. English prompts

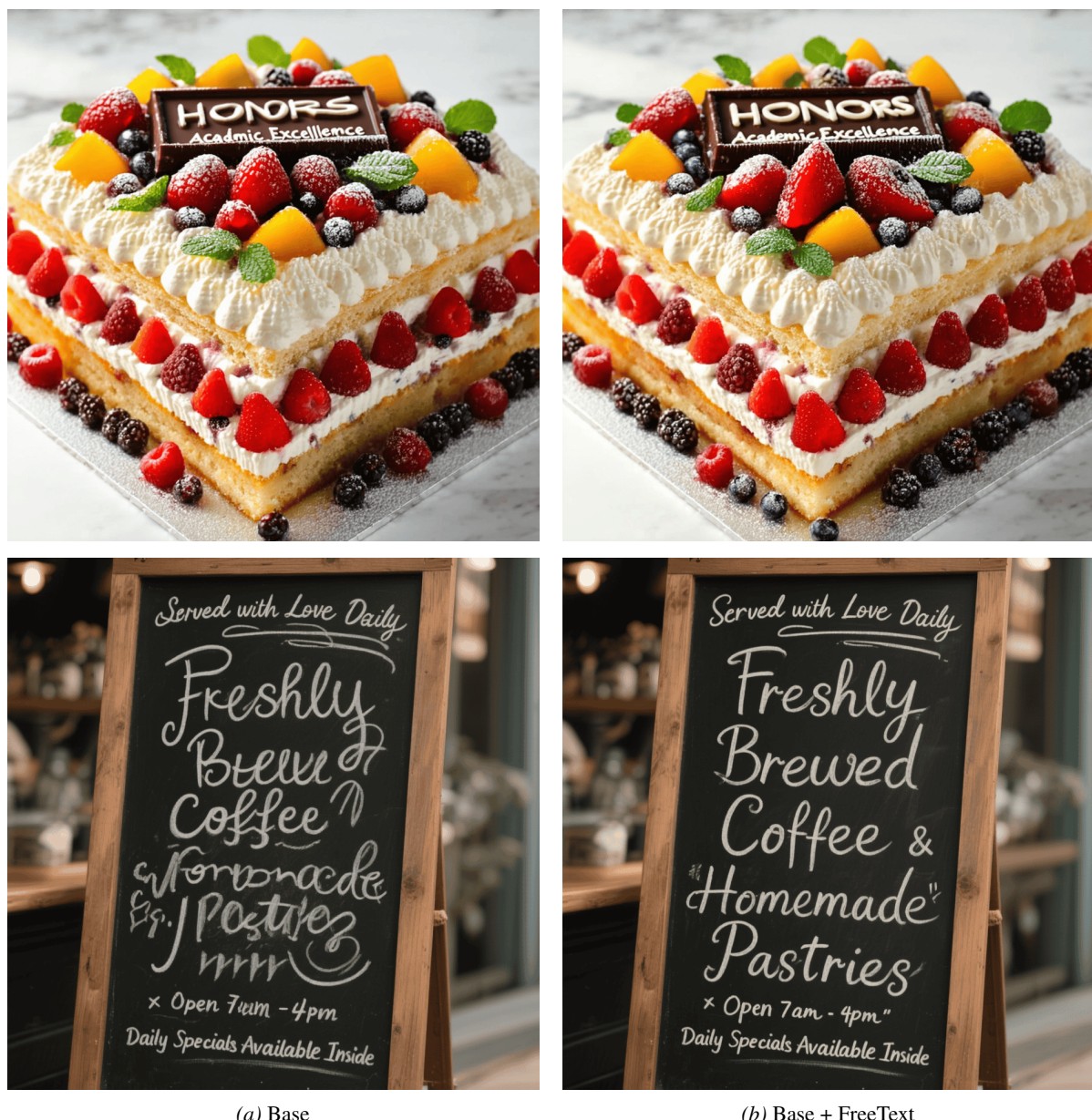

*(a)* Base       *(b)* Base + FreeText

*Figure 10.* English qualitative comparisons. FreeText enhances dense/multi-line text readability while preserving aesthetics.

## A.3. Additional Qualitative Examples with Chinese Prompts

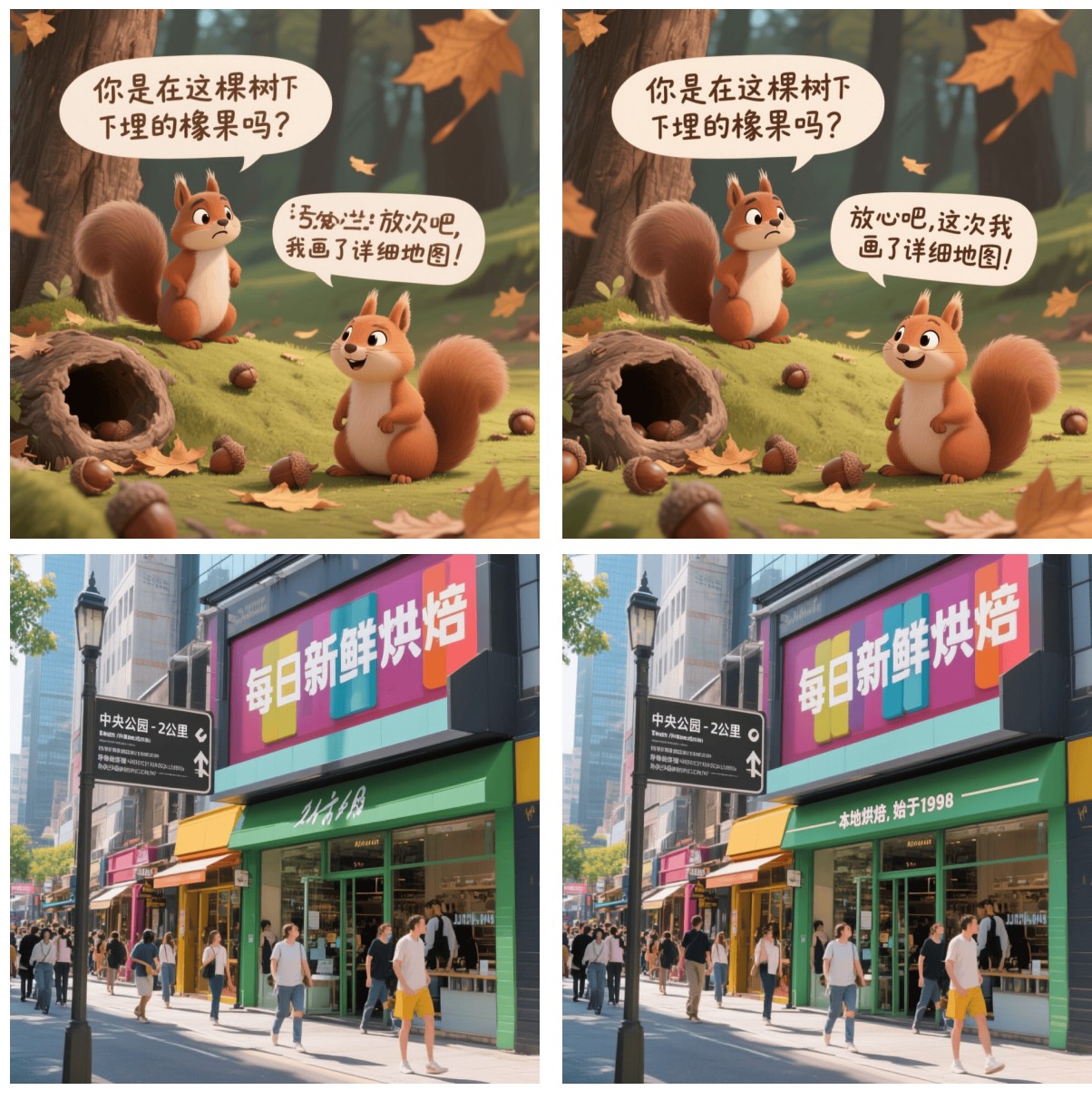

(a) Base            (b) Base + FreeText

*Figure 11.* Chinese qualitative comparisons. FreeText improves long-tailed character fidelity and stroke topology with minimal aesthetic degradation.

## B. Endogenous Priors in Evaluation: Why VLMs Overestimate Text Rendering Quality

Evaluating the typographic fidelity of text-to-image (T2I) models presents a unique challenge. While T2I models often generate text with correct semantic intent, the rendered glyphs frequently suffer from structural ambiguity or "glyph collapse." A common practice is to employ large Vision-Language Models (VLMs) to extract text for metric calculation (e.g., Normalized Edit Distance, NED). However, we argue that this approach introduces a systemic bias, yielding inflated scores that mask actual visual defects.

As revealed in recent investigations into vision-text architectures , end-to-end models—including state-of-the-art VLMs—exhibit a strong "linguistic crutch" phenomenon. Unlike traditional pipeline OCR systems (e.g., PaddleOCR)

that explicitly decouple visual detection from linguistic recognition , VLMs rely heavily on endogenous language priors. When presented with the malformed or ambiguous characters typical of generated images, the VLM's decoder acts as an unintentional "auto-corrector." It tends to hallucinate the correct text based on global context and probability distributions rather than faithfully transcribing the visual evidence.

This discrepancy is quantitatively illustrated in Figure 12. When evaluating generated samples with structural flaws, the VLM (Qwen3-VL-Plus) leverages its priors to reconstruct the intended text, assigning disproportionately high NED scores (e.g., 0.9247 and 1.0). In contrast, a robust pipeline OCR—serving as a proxy for "visual truth" due to its lack of generative priors —correctly identifies the rendering errors, resulting in significantly lower fidelity scores ( 0.77 and 0.60). Consequently, we conclude that VLM-based evaluation predominantly measures semantic recoverability rather than typographic precision, rendering it unsuitable for fine-grained text rendering assessment.

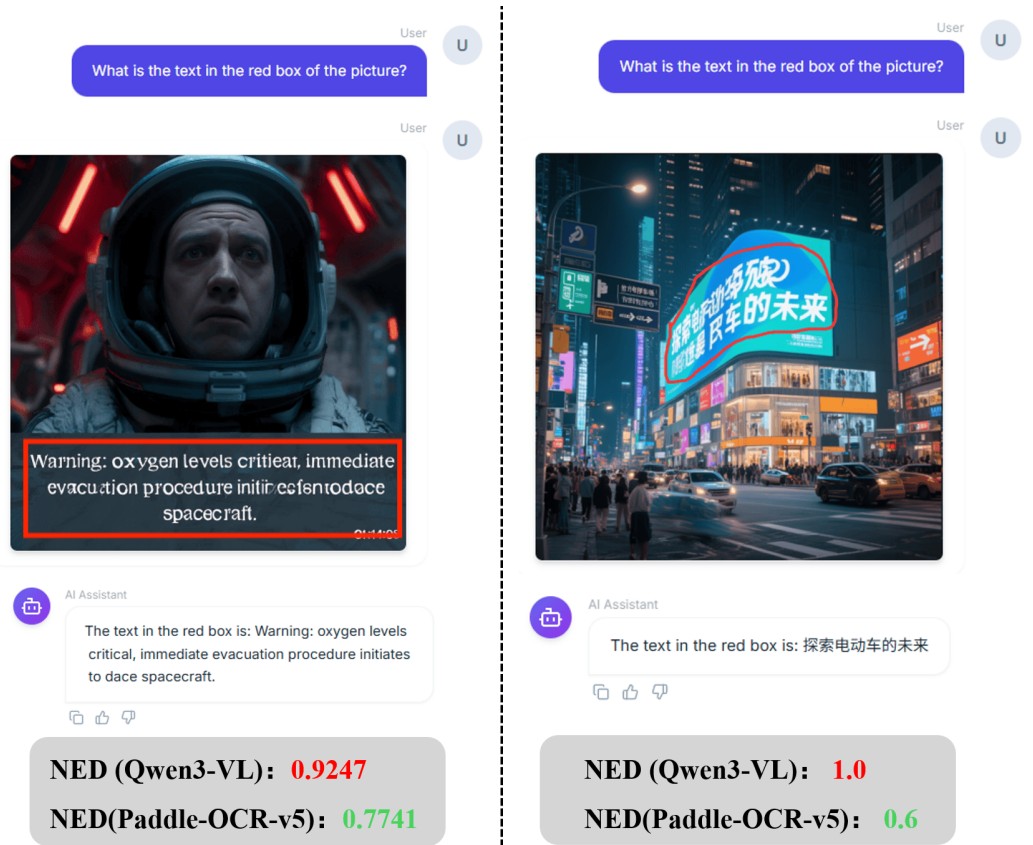

*Figure 12.* The "Linguistic Crutch" effect in VLM-based evaluation. We compare the OCR extraction performance of an end-to-end VLM (Qwen3-VL-Plus) against a traditional pipeline OCR (PaddleOCR) on generated text. The VLM utilizes strong internal language priors to "auto-correct" visually malformed glyphs, resulting in inflated Normalized Edit Distance (NED) scores (e.g., 1.0 vs. the actual visual fidelity of 0.6). In contrast, the pipeline approach provides a more objective assessment of visual defects, confirming that VLM-based metrics often reflect linguistic plausibility rather than visual merit.

## C. VQA Evaluation Prompts

As mentioned in the main text, we employ a VLM-based evaluator to assess image usability and clarity. Below is the full system prompt used for the **VQA Score**.

**System Prompt for VQA Score Evaluation**

You are a hyper-critical quality assurance inspector for a text-to-image generation benchmark. Your task is to evaluate images with forensic, microscopic scrutiny. Your primary directive is to penalize any deviation from physical, anatomical, and logical coherence, unless such deviations are explicitly requested by the text prompt. Assume all subjects and environments must be perfectly sound and plausible by default.

**Scoring System:** You will start with a perfect score of 10 and deduct points for any flaws you identify. A single significant flaw should prevent a high score.

**Flaw Categories (Deduct points for each instance):**

**Critical Failures (-7 to -9 points):**

- Any violation of the fundamental anatomical or structural integrity of the main subjects. This includes inconsistencies in form, function, or natural appearance.
- A breakdown in logical or physical plausibility within the scene, when not specified by the prompt.
- Prominent, distracting digital artifacts, watermarks, or signatures that ruin immersion.
- The central subject is rendered as grotesque or nonsensical, when not specified by the prompt.

**Significant Flaws (-4 to -6 points):**

- Noticeable warping, distortion, or a lack of convincing texture on key objects or surfaces.
- Unnatural blending, texture repetition, or other clear indicators of AI synthesis that break realism.
- Lack of sharpness or resolution in the primary subject, making crucial details indistinct.
- Incoherent or illogical features on secondary elements.

**Minor Imperfections (-1 to -3 points):**

- Slight compositional awkwardness or minor issues with lighting and shadow that don't break realism.
- Minimal blurriness or noise in secondary, non-focal areas of the image.
- Faint, non-distracting artifacts that are only visible upon close inspection.

**Required Output Format:** Your response must be a single JSON object containing a one-sentence "justification" for point deductions and a "score":

```
{{
"justification": ...,
"score": ...,
}}
```

**text prompt:** {text_prompt}

## D. System Prompt for VLM-based Localization

We utilize a commercial multimodal VLM to perform baseline text localization. The following system prompt is explicitly utilized in the section **Comparison with VLM-based localization** (see Section **??** in the main text) to evaluate the baseline VLM performance.

**System Prompt for VLM Localization Baseline**

**Role**
You are a multimodal VLM. Given a text-to-image generation prompt (t2i prompt) and its generated image, you perform fine-grained text alignment: extract all textual content that must appear in the image according to the prompt, locate the corresponding text regions in the image, and provide normalized content and location descriptions.

**Inputs**

- **prompt**: A t2i generation prompt, which may be in Chinese, English, or a mixture of both.
- **image**: An image generated from this prompt (resolution is unconstrained; text may be rotated, skewed, distorted, or partially missing).

**Objective**

- Parse every piece of text that the prompt explicitly requires to be rendered in the image (line by line / segment by segment as clear text).

- For each required text, find the best-matching region in the image; if the text does not appear, explicitly mark it as "not present".
- Output structured JSON, containing only the required fields; do not add any extra or irrelevant keys.

**Reasoning Guidelines**

*Text extraction:*

- Identify text that the prompt explicitly requires to appear in the image (including punctuation and case).
- Prioritize segments inside quotation marks and segments triggered by verbs such as "reads / states / says / labeled as / displays as / titled as", or their equivalents.

*Multilingual:*

- Detect the language type of each text segment. language ∈ {en, zh, mixed, other}.

*Localization strategy:*

- Use layout / positional hints in the prompt (e.g., "top-left corner", "below the title") as semantic priors.
- Perform visual / OCR-style search in the image. Allow for character disorder, missing characters, extra characters; use relative position and overall shape to help match.
- If the same target text has multiple candidate regions, prefer: position semantics match → string proximity → visual clarity.

*Missing text:*

- If the image does not contain the specified text, or only contains unusable fragments, mark it as "not present".

*Attribute inference:*

- Optionally infer text color, font class, style, and emotion. If unsure, use null.

*Position labels & Polygon region (bbox):*

- `position.absolute`: Use nine-grid labels (UL, UC, UR, ML, MC, MR, LL, LC, LR).
- `bbox`: Output a polygon (array of 2D points normalized to [0,1]) tightly enclosing the text. Use at least 3 points, clockwise order. Example: `[[x1,y1],[x2,y2],...,[xn,yn]]`.
- If text is not present, set `bbox: null`.

**Output Format (JSON only)**

Output a single top-level object containing a `texts` array; do not output any explanatory text.

```
{
  "texts": [
    {
      "text": "<exact string>",
      "language": "en|zh|mixed|other",
      "position": {
        "absolute": "UL|UC|UR|ML|MC|MR|LL|LC|LR|null",
        "relative": "<concise anchor phrase or null>"
      },
      "bbox": [[x1,y1],[x2,y2],...,[xn,yn] or null],
      "attributes": {
        "color": "<named or #RRGGBB or null>",
        "font": "<family/class e.g., serif, sans or null>",
        "style": "<e.g., bold, italic or null>",
        "emotion": "<e.g., elegant, playful or null>"
      },
      "reason": "<short explanation; 'not found' if missing>"
    }
  ]
}
```

**Special Cases & Output Rules**

- **Missing text:** Set `bbox: null`, positions to `null`. `reason` must include "not found".
- **Multiple lines:** Output each piece of information as an independent object.
- **Disambiguation:** Briefly describe tolerance basis in `reason` (e.g., "minor character disorder").
- Output only the JSON structure. Do not fabricate details.

