# OpenReview forum: "FreeText: Training-Free Text Rendering via Attention Localization and Spectral Glyph Injection"
_ICML.cc/2026/Conference — ICML 2026 regular_

### Official Review · Reviewer_cBpz · 2026-02-23

**Soundness:** 3
**Presentation:** 3
**Significance:** 3
**Originality:** 3
**Overall Recommendation:** 4
**Confidence:** 3

**Summary:**

This paper addresses a persistent and practically important limitation in diffusion-based text-to-image models: unreliable and low-fidelity text rendering, especially in multi-line and long-form scenarios.
The authors propose a fully training-free inference-time framework that improves text rendering by (1) attention-guided spatial localization of text regions (“WHERE”) and (2) frequency-domain glyph injection for fidelity enhancement (“WHAT”).
The method is plug-and-play and does not require retraining or architectural modification. It is evaluated on multiple large-scale diffusion models and benchmarks, showing improvements in text readability metrics with moderate inference overhead.

**Compliance With Llm Reviewing Policy:**

Affirmed.

**Final Justification:**

My concerns have been fully addressed. Now I would stay with this score. The human evaluation authors provide is not rigid, with only 30 data for human evaluation. Actually with the help of human annotator websites, one day could get around 2k-4k data for evaluation. I would stay with 4.

**Key Questions For Authors:**

1. How sensitive is the method to diffusion scheduler choice and CFG scale?
If robust, this significantly strengthens deployability claims.

2. How sensitive is the injection window?

3. Does the method generalize to models trained with different attention architectures?

4. The solution depends on stable internal attention maps. Can this apply to weaker models, such as UniDiffuser or other relevant models?

**Limitations:**

yes

**Strengths And Weaknesses:**

##  Strengths
### 1. Soundness
The method is technically coherent and clearly implemented at inference time without retraining. They do experiments on different MMDiT-based models, and get consistent gains across different models.
### 2. Presentation
The paper  is clearly structured around the “WHERE” and “WHAT” decomposition, which is intuitive and easy to understand.
### 3. Significance
Text rendering problem is a well-known problem in T2I models. Because T2I models' visual generation ability is getting stronger and stronger, but text generation is still not well addressed. This paper propose a inference time method to tackle it. It would save a great cost, due to the heavy training cost of new T2I models.
### 4. Originality
The use of endogenous attention maps for localization is practical. Frequency-domain modulation for glyph preservation is an interesting engineering innovation.

## Weaknesses
I am not familiar with this area. The paper has many equations that do not need to be presented as equations. Maybe putting it into the appendix or inline equations could make it better for reading.

Another thing is that it's not clearly stated why only choose 0.8-0.6T as the injection window. Also no ablation on this.

---

> ### Author Rebuttal · Authors · 2026-03-30
>
> Thank you very much for the helpful suggestions. In the revision, we will convert non-essential display equations into inline equations and move some technical details to the appendix to improve readability. We also agree that the motivation for the **0.8T-0.6T** **injection window** is currently under-explained, and we will clarify the rationale and add the corresponding ablation.
>
> **1. Regarding sensitivity to diffusion scheduler choice and CFG scale**
>
> The current version does not yet include a systematic ablation over **scheduler** and **CFG scale**, so we do not want to overclaim. That said, from a mechanism perspective, **FreeText** should have a certain degree of robustness to both factors.
>
> For **CFG**, our early observations suggest that overly low CFG makes the attention between **text tokens** and **img patches** more diffuse, which is less favorable for text-region localization. However, **FreeText** does not rely on the attention from a single **timestep / layer**. Instead, it uses **timestep-layer selection** to identify more stable attention patterns, together with **sink-like tokens** as stable anchors. Therefore, it should remain reasonably stable within a practical CFG range.
>
> For **scheduler**, different choices mainly alter the sampling distribution over **timesteps**, while the functional roles of different denoising stages are often relatively consistent: **early steps** are more related to **global planning** and coarse structure generation, **mid steps** are more suitable for forming text-related structural information, and **late steps** are more associated with high-frequency texture and detail refinement. Related analyses of stage-wise and frequency-wise behaviors in diffusion processes can also be found in the following works. Our **SGMI** performs local injection only after alignment to the current **noise level**, and only within a mid-early window, to strengthen glyph structure and reduce structural errors. Therefore, from a mechanism perspective, it should not be overly sensitive to the specific scheduler form.  We will add more direct ablations in the revision to further verify this point.
>
> **Related works:**
>
> - *Beta Sampling is All You Need: Efficient Image Generation Strategy for Diffusion Models using Stepwise Spectral Analysis*
> - *Boosting Diffusion Models with Moving Average Sampling in Frequency Domain*
> - *DMFFT: improving the generation quality of diffusion models using fast Fourier transform*
>
>
>
> **2. Regarding sensitivity of the injection window**
>
> We choose **0.8T-0.6T** because glyph-prior injection should be neither too early nor too late. As already motivated in the paper, this window avoids disturbing early-stage **global planning** while still allowing correction before **glyph structure** becomes hard to modify. Combined with Fig. 3, **early steps** still have coarse attention, whereas **late steps** are more dominated by refinement; thus the mid-early stage is a better compromise between structure formation and controllable correction. We agree that the current version lacks a dedicated window ablation, and we will add comparisons across different **injection windows** in the revision.
>
> **3. Regarding generalization to models with different attention architectures**
>
> The **WHERE** stage of **FreeText** directly reads **image-to-text cross-attention**, so it is most naturally applicable to T2I models with accessible **DiT/MMDiT-style** attention signals. The current paper already shows consistent gains on **Qwen-Image**, **FLUX.1-dev**, **SD3.5-L**, and **SD3-M**, indicating that the method does not depend on a single base model. More generally, the idea should transfer as long as a model provides accessible and reasonably stable **image-to-text cross-attention**; if the architecture differs substantially or stable attention maps are not available, extra adaptation would be needed. We will clarify this applicability boundary in the revision.
>
> **4. Regarding weaker models such as UniDiffuser**
>
> We have not yet evaluated **FreeText** on **UniDiffuser** or other weaker models, so we do not want to make claims beyond the current evidence. Based on our results, **FreeText** mainly strengthens an existing **glyph prior** rather than enabling text rendering capability from scratch. This is also consistent with the observation on **CLT-Bench** that the method is more effective when the base model already has a basic prior for the target characters. Therefore, if a weaker model still provides usable **cross-attention** signals and a basic text prior, **FreeText** may still bring gains; however, the improvement may be less pronounced than on the stronger foundation models studied here.
>
> ---
> **We sincerely thank you again for your valuable time and constructive comments.**

---

> > ### Author Rebuttal · Reviewer_cBpz · 2026-04-01
> >
> > My concerns have been fully addressed. Now I would stay with this score. If the author could provide rigid human evaluation instead of VLM evaluation during the rebuttal period, I will raise my score.

---

> > > ### Author Response · Authors · 2026-04-04
> > >
> > > Thank you very much for your response. We have adopted your constructive suggestion and added the experiment below.
> > >
> > > ---
> > >
> > > ## Pairwise Human Evaluation
> > >
> > > To complement the automatic evaluation results and mitigate the language-prior bias potentially introduced by VLM-based evaluation, we followed your suggestion and designed a **blind pairwise A/B comparison**.
> > >
> > > ### Evaluation Setup
> > >
> > > We followed the same generation protocol as in the main experiments and generated two images for each prompt under identical inference settings: **Base** and **Base + FreeText**. This setting is consistent with the unified “Base vs. Base + FreeText” comparison used in the main experiments.
> > >
> > > To cover typical text-rendering challenges, we sampled from the following three benchmarks: **longText-en**、**longText-zh**、**CVTG**. Specifically, we **randomly sampled 15 prompts** from each benchmark, yielding **45 A/B image pairs** in total. In this human evaluation, `longText-zh` was evaluated by a Chinese annotator pool (via Baidu CrowdTest), while `longText-en` and `CVTG` were evaluated by an English annotator pool (via MTurk); For each sample, the left-right positions of A and B were randomly shuffled, and the method names were hidden from annotators to avoid method priors and position bias. All annotators were also not involved in method development.
> > >
> > > ### Annotators and Quality Control
> > >
> > > We collected **30 questionnaires** in total. To ensure annotation quality, each questionnaire included two types of quality control items:
> > >
> > > 1. **attention check**: a small number of control samples with obvious differences were inserted to filter out careless responses;
> > > 2. **consistency check**: a small number of repeated sample pairs were included to measure intra-annotator consistency.
> > >
> > > Based on the results of the attention check and consistency check, we discarded **4** invalid questionnaires and retained **26 valid questionnaires** for statistical analysis.
> > >
> > > ### Questionnaire Design
> > >
> > > Each evaluation item presented two images corresponding to the same prompt (Image A / Image B), together with the **target text spans explicitly required by the prompt** and necessary layout hints, rather than relying only on open-ended semantic descriptions. Annotators were asked to make pairwise judgments along the following three dimensions:
> > >
> > > | Dimension                  | Question                                                     | Options                                       |
> > > | -------------------------- | ------------------------------------------------------------ | --------------------------------------------- |
> > > | **Text fidelity**          | Which image better renders the target text correctly and legibly? | **A clearly better / B clearly better / Tie** |
> > > | **Layout fidelity**        | Which image better preserves the intended text layout and placement? | **A clearly better / B clearly better / Tie** |
> > > | **Overall visual quality** | Which image has better overall visual quality, ignoring text correctness? | **A clearly better / B clearly better / Tie** |
> > >
> > > ### Aggregation and Reporting
> > >
> > > For statistical analysis, we aggregated the voting results from valid questionnaires into **Win / Tie / Lose** ratios for each dimension, where:
> > >
> > > - **Win**: Base + FreeText is preferred over Base;
> > > - **Lose**: Base is preferred over Base + FreeText;
> > > - **Tie**: no significant difference between the two.
> > >
> > > Based on this, we report the following results:
> > >
> > > - **Win / Tie / Lose (%)** on each benchmark subset;
> > > - the overall preference ratios across all samples.
> > >
> > > **Per-subset results**
> > >
> > > | Subset      | Text fidelity (W/T/L, %) | Layout fidelity (W/T/L, %) | Overall visual quality (W/T/L, %) |
> > > | ----------- | ------------------------ | -------------------------- | --------------------------------- |
> > > | longText-en | 65.2 / 18.1 / 16.7       | 62.9 / 20.5 / 16.7         | 24.3 / 60.5 / 15.2                |
> > > | longText-zh | 61.7 / 22.8 / 15.6       | 57.2 / 23.3 / 19.4         | 20.0 / 66.1 / 13.9                |
> > > | CVTG        | 59.0 / 22.4 / 18.6       | 60.5 / 23.8 / 15.7         | 18.1 / 71.9 / 10.0                |
> > >
> > > **Overall results**
> > >
> > > | Evaluation Set | Text fidelity (W/T/L, %) | Layout fidelity (W/T/L, %) | Overall visual quality (W/T/L, %) |
> > > | -------------- | ------------------------ | -------------------------- | --------------------------------- |
> > > | All subsets    | 62.0 / 21.0 / 17.0       | 60.3 / 22.5 / 17.2         | 20.8 / 66.2 / 13.0                |
> > >
> > > Overall, the **Pairwise Human Evaluation** further shows that FreeText can more consistently improve text legibility and content fidelity from the perspective of human perception, and still exhibits a consistent preference advantage under direct human comparison; meanwhile, the dominance of **Tie** in **Overall visual quality** also indicates that the method does not significantly compromise the naturalness or aesthetic quality of the overall image while enhancing text rendering.

---

### Official Review · Reviewer_xPHX · 2026-03-07

**Soundness:** 3
**Presentation:** 2
**Significance:** 2
**Originality:** 3
**Overall Recommendation:** 4
**Confidence:** 3

**Summary:**

This article outlines a major issue that existing text-to-image diffusion models perform poorly in precise text rendering and prior methods rely on costly retraining or rigid layout constraints. Overall, the article presents a central concept of decomposing text rendering into “where to write” and “what to write” to build a training-free plug-and-play framework called FreeText.

FreeText locates text regions using attention localization with sink-like tokens and enhances glyph fidelity via Spectral-Modulated Glyph Injection (SGMI) to suppress semantic leakage. The authors also contribute CLT-Bench, a dedicated benchmark for evaluating long-tailed Chinese text rendering. Experiments on multiple mainstream models verify that FreeText boosts text readability while keeping semantic alignment and aesthetics with small inference overhead.

**Compliance With Llm Reviewing Policy:**

Affirmed.

**Final Justification:**

My concerns have been solved and I will maintain my rating.

**Key Questions For Authors:**

see weaknesses

**Strengths And Weaknesses:**

**Strengths:**

1. This article outlines a major issue in modern text-to-image models: poor text rendering (especially for long-tailed Chinese and multiline text) caused by costly retraining or rigid layout constraints, and the paper effectively solves it with a training‑free plug‑and‑play framework.

2. Overall, the article presents a central concept: decomposing text rendering into “where to write” (attention‑guided localization) and “what to write” (spectral glyph injection), which is intuitive, novel, and widely applicable to DiT‑based models.

**Weaknesses:**

1. In the related works, the authors already referred to prior method including: TextDiffuser-style, AnyText, GlyphDraw, GlyphControl, and UniGlyph. Although we already know that they suffer from overhead and extra training, we should still compare with some of them to prove that your method could work better than these specific glyph text generation methods.

2. The spectral modulation parameters (e.g., Log‑Gabor kernel settings) are manually predefined without adaptive learning, making the method less robust across diverse fonts, styles, and languages.

---

> ### Author Rebuttal · Authors · 2026-03-30
>
> **1. Regarding direct comparisons with TextDiffuser-style / AnyText / GlyphDraw / GlyphControl / UniGlyph**
>
> Thank you for the suggestion. We agree that direct comparison with existing text-specialized methods is valuable. We did not include them in the main experiments for four main reasons:
>
> 1. **The model scales are not comparable, so a direct comparison is not fully fair.**
>     This work targets large T2I foundation models with strong **open-domain generation quality**, such as **FLUX.1-dev** and **Qwen-Image**. **FreeText** is designed as a **training-free** enhancement on top of such models, rather than a separately retrained text generation system.
> 2. **Methods such as UniGlyph and AnyText typically rely on CLIP tokenizers, which are disadvantaged in long-prompt settings due to truncation.**
>     Our benchmarks explicitly include **longText-Benchmark**, which focuses on **long prompts** and **paragraph-level, multi-line text**. Under such settings, CLIP-tokenizer-based methods are difficult to align strictly with our evaluation, and may also be disadvantaged in **CVTG** cases requiring longer textual specifications.
> 3. **Existing public results already show a clear gap from current large foundation models on relevant benchmarks.**
>     For example, **Table 2 of TextCrafter** shows that **AnyText** and **TextDiffuser-2** are clearly behind **SD3**, **FLUX.1**, and **Qwen-Image** on **CVTG-2K** in **Word Accuracy**, **NED**, and **Aesthetics**. As a result, using them as the main baselines here would add limited information, since they are no longer in the same performance regime as current large-scale foundation models.
> 4. **More importantly, the two lines of work differ fundamentally in problem setting.**
>     As summarized in our **Related Work**, methods such as **TextDiffuser-style** and **AnyText** mostly follow a **retraining-based**, **layout-dependent** paradigm, requiring extra **box / mask / glyph / segmentation** inputs. In contrast, **FreeText** is a **training-free, plug-and-play** **inference-time enhancement** that improves existing large-scale T2I foundation models without changing model parameters or architectures and without requiring **rigid layout conditions**. Thus, the two categories differ substantially in user input, control mechanism, and method objective.
>
> ------
>
> **2. Regarding manually predefined spectral modulation parameters**
>
> Thank you for pointing this out. We indeed do not use adaptive learning for the **Log-Gabor kernel** parameters. However, these parameters are not chosen arbitrarily; they are stable settings obtained through extensive empirical analysis and parameter search.
>
> From the method perspective, **Spectral-Modulated Glyph Injection (SGMI)** addresses **WHAT to write** by strengthening glyph structure while suppressing **semantic leakage**. Its purpose is not to inject more information, but to preserve the spectral band most beneficial to glyph structure, namely the **mid-to-high frequency components that carry glyph structures**, while suppressing low-frequency background and irrelevant noise.
>
> Our observations during parameter search were consistent: retaining too much low-frequency content leads to results like Fig. 8(c), where glyph structure is weakened and background dominates; overly strong high-frequency content more easily causes the **concept-texture intrusion** shown in Fig. 8(d), i.e., stronger **semantic leakage**. The current band-pass setting provides the best balance between glyph fidelity and semantic suppression. This is also supported by the ablation: removing **SGMI** clearly reduces **NED** and **VQA Score**, while **CLIPScore** and **AestheticScore** remain largely stable, indicating that SGMI mainly improves text readability rather than benefiting from incidental parameter fitting.
>
> More importantly, this setting is not effective only for one model or one benchmark. Across **Qwen-Image**, **FLUX.1-dev**, **SD3.5-L**, and **SD3-M**, as well as **longText-Benchmark**, **CVTG**, and **CLT-Bench**, **FreeText** consistently improves readability while largely preserving semantic alignment and aesthetics. This suggests that the current spectral modulation setting already has reasonable robustness across architectures and scenarios.
>
> We nevertheless agree that **adaptive learning** or sample-wise adaptive spectral modulation is a meaningful future direction. However, when T2I text rendering fails, the forward process often does not first produce a stable and correct glyph structure; under this condition, relying too early on adaptive estimation may introduce extra instability. For this reason, in the current version we prioritize a thoroughly validated fixed setting that is relatively robust across models, and we will explore more reliable adaptive variants in future work.
>
> ---
> **We sincerely thank you again for your valuable time and constructive comments.**

---

> > ### Author Rebuttal · Reviewer_xPHX · 2026-04-03
> >
> > My concerns are resolved.

---

### Official Review · Reviewer_D5M7 · 2026-03-12

**Soundness:** 1
**Presentation:** 2
**Significance:** 3
**Originality:** 3
**Overall Recommendation:** 2
**Confidence:** 5

**Summary:**

The paper presents FreeText, a method for improving the ability of pretrained image generation models to render text without fine-tuning and without imposing external constraints or layout supervision, by using the model’s attention maps to locate the regions of interest, and by using a band-pass filter in some stages of generation to only allow the specific frequency bands associated with glyphs to appear in the generation process within those regions of interest. It additionally proposes a benchmark for Chinese long-tail text rendering. The localization component and band-pass filter for generation are also listed as individual contributions. The method is evaluated in comparison to its own baseline architecture. Results show that the proposed method improves text rendering performance of four tested architectures (FLUX, Qwen-Image, and two different Stable Diffusion 3 variants), with a moderate efficiency overhead. The attention map-based localization method is compared to a VLM prompting approach, tested on four different VLMs, and, in reported results, it exhibits higher Intersection over Union scores than all of them.

**Compliance With Llm Reviewing Policy:**

Affirmed.

**Final Justification:**

I thank the authors for engaging in the discussion. Their responses partially clarified some of the points in my review and restated the authors’ previous arguments. Nonetheless, these did not address some key outstanding concerns.

Specifically, despite the paper’s claims about superiority with respect to existing methods being abundant, they are not backed by experimental results. In any case, even if such claims were to be toned down, I still strongly believe that such comparisons are necessary to contextualize the contribution with respect to the state of the art to tackle a given task (especially one that is challenging, timely, and underexplored, such as the one at hand). This holds regardless of the fact that the methods themselves are different from each other and from the proposed one.

Moreover, I’m afraid the authors misused the space at their disposal by lengthily restating claims and, as a result, they did not provide information about the proposed benchmark, which is listed as a key contribution of this paper. Without such information, the extent of this contribution cannot be validated (nor even assessed) at revision time.

As such, I’m afraid my assessment is not substantially altered by the authors’ responses. I still judge this paper to be lacking in terms of soundness and presentation. Even though the proposed method itself has some merits, the paper is unfit for publication as it stands.

**Key Questions For Authors:**

1. How does the model compare to existing approaches for text rendering in the literature, like AnyText, TextDiffuser, or GlyphByT5?
2. Is the NED score you reported the same as that used by AnyText?
3. Does your text rendering framework require two generation runs? Is this taken into account in Table 7? If so, how can the generation process not take at least twice as much as the baseline generation?
4. Was the model only evaluated by generating one text region at a time?
5. When are the timestep and layer selection performed?

**Limitations:**

The authors did not discuss the limitations of their work, such as the apparent difficulty in applying their method to multi-colored backgrounds. The overall very limited evaluation of their method makes it difficult to even ascertain the strengths and weaknesses of their approach when compared to other existing approaches.

**Strengths And Weaknesses:**

Strengths:

- The method is novel and principled, and the claims about leveraging the model’s endogenous layout planning abilities are strong and relevant to the current state of the art.
- The method is implemented and tested on four different architectures, including FLUX.1-dev, Qwen-Image, and two variants of Stable Diffusion 3, and the authors report incremental score improvements with respect to the baseline on all four architectures
- The proposed benchmark is reported in Table 3 to be more challenging than existing datasets for the proposed method and its Qwen-Image baseline, supporting the claims in Section 1 and Section 3.3 that it highlights weaknesses in long-tail rendering.

Weaknesses:

- The text rendering framework, which is the main contribution, is not experimentally compared to any other text rendering frameworks present in the literature
- Section 1 claims that extensive finetuning, as done in other approaches in literature (e.g., the referenced AnyText, TextDiffuser, GlyphDraw, GlyphControl, and UniGlyph), is detrimental to aesthetic quality. This motivates the approach proposed by the authors. Nevertheless, this claim is not substantiated in the paper (especially, it is not demonstrated experimentally).
I think that it could be argued that also filtering certain frequencies in the regions where text would appear can reduce aesthetic quality, as it would affect the model’s ability to generate texture, only allowing the model to generate solid font-like text on a solid, uniform (or almost completely so) background. Indeed, the qualitative results in Figures 4, 5, 8, 10, and 11 only show this specific case, without showing other cases. This aspect is not addressed anywhere in the paper as a limitation of the method. Moreover, the second row of Figure 11 (in Appendix A) even shows how the model, when applying FreeText, stops generating a sunlight reflection, which would create an uneven background.
- Related to the previous weakness, low-rank finetuning methods, or other region-based approaches that address the mentioned issue on aesthetic quality degradation (e.g., Liu et al., ”Glyph-ByT5: A Customized Text Encoder for Accurate Visual Text Rendering”, ECCV, 2024, and Ma et al., “Calligrapher: Freestyle Text Image Customization”, ACMMM, 2025) are not cited.
- One of the listed contributions in Section 1 is the benchmark on long-tail Chinese text rendering. However, there are no details about the size and content of the dataset, no qualitative samples, no information about the data collection process, no overview of the current literature on text rendering benchmarks in the related work section, nor any testing of current state-of-the-art text rendering models on the proposed benchmark. Section 3.3, which describes the benchmark, only defines a complexity score for a given text prompt and mentions that the dataset is stratified into subsets based on this score. However, there is no information about how many subsets there are, how many samples each contains, or what the ranges used for this stratification are. This complexity score is not used anywhere in the paper, and this stratification is not used nor mentioned in the experimental results presented in Section 4.
- The experimental evaluation of the text localization component is only done against some VLMs prompted with a prompt written by the authors of the paper (and presented in Appendix D). This component should rather be evaluated against approaches used for text detection in the existing literature, for example, the one used in the cited Cui et al. paper presenting PaddleOCR, or any other paper presenting text detection approaches.
- It is unclear whether the localization step is integrated into the generation process, thus only requiring a single generation run, or whether a generation run is required to extract the attention maps and then a second generation run using SGMI is performed. In this second case, which is suggested by the overview in Figure 2, the inference efficiency results reported in Table 7 seem not to take both inference runs into account, since they show a ~10-15% overhead.
- The “sink-like special tokens” mentioned in Section 3.1.1 and added to the textual sequence are never specified in kind and number.
It is unclear whether the layer and timestep selection described in Section 3.1.2 is performed for every inference run, for each architecture, for each architecture-dataset pair, etc. The necessity of reference masks Y also contradicts the claim in Section 2.2 that other models rely on extra annotations, and it is not shown as an input in Figure 2.
- There is no information on the experimental setting used to evaluate the localization component: Section 4 contains two tables (Table 4 and Table 5) presenting IoU results, with no information on which dataset these scores refer to.
- The Normalized Edit Distance (NED) reported in Section 4 as a higher-is-better score, acting as some sort of character accuracy, is not coherent with either any definition of NED in the current and past literature (e.g. A. Marzal and E. Vidal, "Computation of normalized edit distance and applications", TPAMI, 1993 or Yujian and L. Bo, "A Normalized Levenshtein Distance Metric", TPAMI, 2007), nor any intuitive form of normalization of Edit Distance, which, in any case, would express distance and not accuracy/similarity and would therefore have to be interpreted as a lower-is-better score in this context. To be fair, also the cited Tuo et al. paper presenting AnyText reports a higher-is-better score called NED, but the paper does not specify how it is computed. AnyText's repo contains an implementation of NED as well, which is 1-d where d is the Levenshtein distance divided by the length of the longest of the two strings. This could be the NED used in this paper, but it is still unclear.
- The writing needs polishing, and the references should be checked: three published papers (Tuo et al., 2023, Razzhigaev et al., 2025, Kang et al., 2025) have been cited in the arXiv version; the reference Otsu, N. et al. “A threshold selection method from gray-level histograms.”, Automatica, 11(285-296):23-27, 1975 is completely wrong

---

> ### Author Rebuttal · Authors · 2026-03-31
>
> Thank you for the detailed comments. As the key questions largely overlap with the listed weaknesses, we address both together below.
>
> 1. Direct comparison with existing text-specialized methods is valuable, but we did not include them as main baselines because：
>    - FreeText targets training-free enhancement of strong large-scale T2I foundation models without changing model parameters or architectures, whereas methods such as AnyText are relatively small text-specialized models trained specifically for text rendering, with overall capability substantially below current large-scale foundation models, making direct comparison not fully fair.
>    - Methods such as UniGlyph and AnyText typically rely on CLIP tokenizer, which suffers from truncation under long prompts and therefore does not align well with our benchmarks.
>    - Public results already show a clear gap: e.g., Table 2 in TextCrafter shows that AnyText and TextDiffuser-2 are clearly behind SD3, FLUX.1, and Qwen-Image on CVTG-2K in Word Accuracy, NED, and Aesthetics;
>
> 2. Prior work such as DreamBooth and Continual Diffusion has repeatedly shown that task-specific fine-tuning introduces a trade-off: it can damage pretrained priors, causing language drift, reduced diversity, or weaker general-purpose generation/compatibility; this is also why the methods mentioned in your third point explicitly seek to mitigate such side effects. Empirically, however, we do not observe systematic aesthetic degradation from FreeText: across models and benchmarks, CLIPScore and AestheticScore remain largely stable.
>
>    As for the qualitative figures, comic / caption / poster / slide were chosen because they are common and challenging text-rendering scenarios, especially for dense or multi-line text. The cited “sunlight reflection” is not required by the prompt; Local latent intervention can slightly alter the diffusion trajectory, leading to minor changes in local lighting or texture; however, this does not imply systematic image-quality degradation.
>
> 3. We will add the missing citations, including the works you mentioned; at the same time, we emphasize that FreeText addresses this issue from a different angle, as a training-free, plug-and-play method requiring no extra fine-tuning, training data, or new learnable branches.
>
> 4. We agree that the current description of CLT-Bench is insufficient; in the revision, we will add dataset size, subset ranges, sample distribution, qualitative examples, and the construction pipeline, and release the dataset together with its construction code. As for comparisons with state-of-the-art text-rendering models, open-source models that support Chinese text rendering and can be directly evaluated in this setting are still very limited at submission time, and Qwen-Image is the most representative foundation model among them.
>
> 5. Our localization task is prompt-conditioned text-region grounding, not generic scene-text detection; when a prompt contains multiple target text segments, a standard OCR/detector pipeline cannot directly solve the correspondence between each target text string and the correct image region, which is why we used prompt-aware VLM baselines.
>
> 6. No. FreeText is integrated into a single denoising trajectory: Stage 1 reads and selects image-to-text cross-attention during sampling, and Stage 2 injects the glyph prior only within the mid-early window (0.8T-0.6T). The extra cost mainly comes from Stage-1 attention accumulation/selection, not from a second full pass, which is why the measured latency increase is only moderate rather than close to 2x.
>
> 7. We agree that this part is currently under-specified.
>
>    - sink-like special tokens are not introduced ad hoc, but follow prior studies on attention sinks already cited in our Related Work (e.g., Tigges et al., 2023; Razzhigaev et al., 2025), and refer here to a small set of punctuation / structural special tokens that show relatively stable high responses across layers and heads.
>
>    - timestep/layer selection is an offline calibration step performed once per model, not re-run for every inference sample;
>
>    - the reference mask **\(Y\)** is used only for this offline selection / IoU analysis on a small annotated set, and is **not** an inference-time input. Therefore, this does not contradict our claim of zero-layout supervision at inference time. We will clarify these implementation details, including the token choice and calibration protocol, in the revision.
>
> 8. We will explicitly state the source and construction of the IoU evaluation subset in the revision.
>
> 9. Our NED follows the higher-is-better convention adopted in recent text-rendering work, including AnyText and technical reports such as Qwen-Image and LongCat-Image, i.e., it is effectively reported as a normalized similarity form (1 - d).
>
> 10. We will further polish the writing, replace arXiv citations with published versions where applicable, and correct the erroneous reference entries.
>
> ---
> Thank you again

---

> > ### Author Rebuttal · Reviewer_D5M7 · 2026-04-02
> >
> > ## Key Questions For Authors
> > Concerning the questions in my original review, my comments on the authors’ rebuttal are as follows.
> >
> > ### Question 1
> > The authors state their refusal to test their method against any competitors
> > 1. By claiming that they don’t need experimental comparison with models that aren’t also training-free adaptations of existing models.
> > 2. By claiming that some of these models will fail because they encode the text using CLIP, which has a very limited sequence length compared to the length of prompts in the proposed benchmark.
> > 3. By pointing to an arXiv preprint proposing a competing training-free adaptation for text rendering (TextCrafter) that reports that some of the mentioned methods perform worse in some scores than this paper’s baseline on the dataset proposed by that preprint, which is also one of the datasets used to evaluate the method proposed by this paper.
> > 4. By reasserting their claims about their superiority in aesthetic preservation when compared to existing approaches, citing DreamBooth and Continual Diffusion as demonstrations that finetuning alters pretrained priors.
> >
> > I have the following comments:
> > 1. The paper proposes a text rendering framework. Thus, it’s not possible to evaluate its relevance without appropriate comparison to other state-of-the-art text rendering frameworks on existing benchmarks. Further, the paper makes repeated claims about such frameworks in Sections 1 and 2, which should be backed by experiments.
> > 2. This potential limit in the text encoding of some of the existing models in the literature is not a sufficient reason not to test any of the models in the literature. GlyphByT5 even proposes ByT5 as a text encoder specifically tailored for this task, and AnyText does not use CLIP to encode the text to be generated, instead using the embeddings obtained from running an OCR model on image renderings of the given text.
> > 3. The result pointed out by the authors comes from a non-peer-reviewed preprint, proposing another training-free text rendering framework. Since the inference code is available and it was known and cited by the authors (who even used their proposed benchmark), at least this approach could have been used as a competitor.
> > 4. The fact that finetuning alters pretrained priors in ways that can be harmful in some settings does not demonstrate that finetuned models cannot outperform or match their baseline in terms of aesthetic quality, so it is harmful to make this generic claim without experimental backing.
> >
> > ### Question 2
> > This question was answered. Given the confusing naming and inconsistent usage of it in the existing literature, I suggest the authors revise their manuscript to clarify this point. In general, further details about the scores would improve the clarity and reproducibility of the results.
> >
> > ### Question 3
> > Answer 6 somewhat attempts to answer this question, but it is still unclear. When exactly is Stage 1 applied? What is “sampling” if not an extra inference run?
> >
> > ### Question 4
> > The authors did not answer my question 4, which I reiterate here: Was the model only evaluated by generating one text region at a time?
> >
> > ### Question 5
> > The second and third bullet points of answer 7 state that this is part of a hyperparameter tuning step done before running the main experiments. Is it done once for each dataset, or are the same hyperparameters used for all datasets? What layers are used? What is the “small annotation set” used for this step?
> >
> > ## Authors’ rebuttal
> > As for the points from the rebuttal, my comments are as follows.
> > 1. Please see my comments about Question 1
> > 2. Please see my comments about Question 1. I note the reluctance to show any more results that contradict my concerns.
> > 3. Contextualization within the existing literature is necessary even when most of it follows a different approach than the proposed method.
> > 4. Can you provide here the missing information about the proposed benchmark? As for now, it is impossible to assess its relevance and soundness.
> > 5. I argue that it is quite straightforward to set up a baseline localization method that uses text detectors and OCR to match the text and/or use one of the many prompt-guided detection models available in the literature. In my opinion, any of these specialized approaches is more relevant than a general VLM in terms of the significance of the comparison.
> > 6. Please see my comments about Question 3.
> > 7. As for the first bullet point: this is a general description of attention sinks. For clarity and reproducibility, the paper should more clearly state what tokens were used and why. As it stands, the language is too vague and unclear. For the other bullet points, please see my comments about Question 5.
> > 8. Can you provide here the missing information on the experimental setting used to evaluate the localization component in Section 4?
> > 9. Thank you for the clarification.
> > 10. Thank you for your willingness to polish the paper and amend the wrong citations.

---

> > > ### Author Response · Authors · 2026-04-07
> > >
> > > Thank you. Due to space limits, we cannot address every point here, but will include all requested details in the revision.
> > >
> > > Q1:
> > > Our main claim is not that FreeText universally outperforms every text-rendering framework on a single benchmark. The question we study is narrower: whether a training-free, inference-time plug-in can consistently improve strong pretrained foundation models without modifying their parameters, architecture, or adding new learnable branches. For this question, the most direct evidence is the paired gain under matched base-model settings, i.e., Base vs. Base+FreeText, rather than a flat comparison between a specialized standalone generator and a plug-in enhancement for an existing generator.
> > >
> > > More importantly, most prior methods address a different deployment question. Methods such as AnyText and TextDiffuser follow a retraining- and layout-dependent paradigm, requiring extra training, control branches, or external conditions such as glyphs, positions, or masks. By contrast, FreeText is not intended to replace a strong T2I foundation model with another text-specialized system; its goal is to enhance the text rendering ability of the existing model at inference time while preserving its original generative prior. Therefore, such comparisons can be useful as contextualization, but they are not the primary evidence for our central claim.
> > >
> > > Even for the training-free competitor you mentioned, TextCrafter and FreeText still differ in both problem setting and technical bias. TextCrafter emphasizes text-carrier coupling and is closer to a carrier-explicit multi-region CVTG setting. FreeText instead decomposes the problem into WHERE and WHAT: it localizes writing regions from endogenous attention and injects a noise-aligned glyph prior, without requiring an explicit method-level binding between the target text and a carrier. Although both are training-free, they are not interchangeable methods under the same formulation.
> > >
> > > Q2:Thank you. We will revise this part accordingly and clarify the metric definition more explicitly.
> > >
> > > Q3:It is not a second full generation run. SGMI is injected in Stage 2 within 0.8T-0.6T. Before injection, we run roughly an additional 0.15T steps to collect the attention needed by Stage 1, which improves localization accuracy. We do not extend this window further because, after Stage-1 selection, the chosen timesteps are already sufficient for reliable localization. This is a trade-off between accuracy and efficiency.
> > >
> > > Q4:No. All target text regions are processed together, rather than one region at a time. Moreover, the mask corresponds to the glyph/stroke area, not the entire text carrier region.
> > >
> > > Q5:All datasets share one model-level hyperparameter setting. In other words, for each base model, this calibration is done only once. We found the attention pattern of each model to be relatively stable across datasets. The small annotated set consists of 200 manually constructed prompts covering different scenes, lengths, and languages; for each model, we manually annotate the target text regions in the generated images. The full prompts, annotations, and the exact selected layers will be released later; due to the rebuttal space limit, we cannot include all these details here.
> > > Other points:
> > >
> > > 1/2/3/6:We appreciate these comments and will carefully revise the manuscript accordingly.
> > >
> > > 4:Due to the rebuttal space limit, we will add these benchmark and protocol details in the revised version.
> > >
> > > 5:This task is often far from ideal for a detector+OCR pipeline. A single prompt may contain multiple target texts, while the generated image may also contain extra text, missing words, wrong words, or misplaced text, making it difficult to match the detected text regions with the target set required by the prompt. Hungarian-style matching (e.g., LeX-Art) usually assumes that OCR segmentation is broadly consistent with the ground-truth segmentation; once they diverge, matching can become unreliable. For example, the ground truth may be “understanding visual text rendering”, while OCR returns three boxes: “visual text”, “rendering”, and “understanding”. In this case, all words are present, but the segmentation and ordering no longer align with the ground-truth span, so the matching result becomes unreliable. Similar issues also arise when the text content is correct but bound to the wrong carrier or attributes.
> > >
> > > 7:By sink-like special tokens, we do not mean arbitrary sink tokens. We refer to special symbols that are adjacent to the target span, carry little semantic content themselves, but maintain a stable structural relation to the target text, most typically quotation marks or punctuation. For example, in a prompt such as "a sign that reads 'hello'", the quotation marks around 'hello' can serve as anchor tokens: they do not encode the text content itself, but are naturally attached to the target text and therefore align more stably with its spatial region
> > >
> > > 8:Please see our response to Q5

---

### Official Review · Reviewer_WXWT · 2026-03-13

**Soundness:** 3
**Presentation:** 4
**Significance:** 3
**Originality:** 3
**Overall Recommendation:** 4
**Confidence:** 4

**Summary:**

Paper proposes FreeText, a training-free framework for improving text rendering in text-to-image diffusion models. The authors argue that existing approaches either require retraining or rely on rigid layout constraints, which often compromise generation flexibility and aesthetic quality. Instead, the paper decomposes the problem into two sub-tasks: where to write and what to write. To address where to write, the method extracts cross-attention maps from diffusion transformers and leverages sink-like tokens as spatial anchors to localize candidate text regions without external supervision. To address what to write, the paper introduces Spectral-Modulated Glyph Injection (SGMI), which injects glyph priors in the frequency domain using Log-Gabor filtering to enhance structure-carrying frequencies while suppressing semantic leakage. The approach is implemented as an inference-time plug-in that does not modify model weights or architecture. Experiments on several benchmarks (LongText-Benchmark, CVTG, and the newly proposed CLT-Bench) show consistent improvements in OCR-based metrics such as NED and VQA-based readability scores while maintaining comparable CLIP and aesthetic scores.

**Compliance With Llm Reviewing Policy:**

Affirmed.

**Final Justification:**

Authors have addressed all my concerns, thus I raise my rating.

**Key Questions For Authors:**

1.How does the method perform under extremely dense or overlapping text layouts? Additional quantitative analysis on such cases would strengthen the claims.

2.The experiments mainly focus on English and Chinese. Could the authors comment on how the method would generalize to scripts with different structural properties (e.g., Arabic, Devanagari)?

3.Given the potential bias of VLM-based evaluation discussed in the appendix, have the authors considered conducting human readability studies or OCR-based structural metrics to complement the evaluation?

4.Could the authors provide more details about CLT-Bench construction and whether the dataset will be publicly released?

**Limitations:**

Partially. The paper discusses biases in VLM-based evaluation and highlights potential limitations of existing evaluation pipelines. However, the discussion of other limitations (e.g., applicability to diverse languages or extremely complex layouts) could be expanded.

**Strengths And Weaknesses:**

This paper addresses the long-standing challenge of text rendering in diffusion-based text-to-image models, particularly under complex layouts and long-tail character distributions. The proposed approach is appealing in that it does not require retraining or architectural modification, and instead leverages intrinsic mechanisms already present in diffusion transformers.

---

> ### Author Rebuttal · Authors · 2026-03-30
>
> Thank you for the constructive comments. Our responses are as follows.
>
> **1. Regarding extremely dense or overlapping text layouts**
>
> We agree this is an important robustness test. Our experiments already target **multi-line layouts**, **dense typography**, and **paragraph-level, multi-line text**, and systematically evaluate **longText-Benchmark** and **CVTG**, which cover long multi-line and multi-region text rendering. Fig. 4 also includes dense cases such as **caption / poster / slide**. Quantitatively, on **Qwen-Image**, **longText-en** improves from **0.625 / 2.650** to **0.713 / 4.177** in **NED / VQA Score**, and **longText-zh** from **0.639 / 3.657** to **0.694 / 4.211**. Sec. 4.2.4 further shows **cross-region benefit propagation**: under global self-attention in **DiT/MMDiT**, improving one key text region can reduce interference to others, which is particularly helpful in dense scenes. We agree, however, that the current version does not isolate a dedicated **severely overlapping text** subset; in the revision, we will add a finer-grained case-based quantitative breakdown.
>
> **2. Regarding generalization to scripts such as Arabic and Devanagari**
>
> First, the tokenizers of most current open-source T2I models do not support scripts such as Arabic and Devanagari. Second, **FreeText** itself is not intrinsically tied to Chinese or English: it decomposes text rendering into **WHERE to write** and **WHAT to write**, using **image-to-text attention** for localization and **SGMI** to inject glyph priors. From a design perspective, it is therefore a **training-free**, **model-agnostic** **inference-time plug-in**. That said, its role is to strengthen an existing glyph prior rather than enable a model to generate a script it has not truly mastered from scratch. As stated in Sec. 4.2.3, **FreeText** is most effective when the **base model already has a basic prior for the target characters**. Therefore, for scripts such as Arabic or Devanagari, we expect it to help when the base T2I model already has some script prior. We will clarify this boundary and the corresponding future validation plan in the revision.
>
> **3. Regarding VLM-based evaluation bias and possible human studies / OCR-based structural metrics**
>
> We fully agree that one should not rely on a single VLM-based metric. In fact, building a more objective and effective evaluation framework for text rendering in text-to-image generation is one of our ongoing efforts, and we plan to release it publicly in the near future.
>
> Methodologically, the new framework adopts a cautious **cross-model validation** strategy: **OCR/text spotting** serves as the primary measure of objectively visible textual facts, while **VLM/QA** is introduced for complementary verification and attribute evaluation (e.g., **VQAScore**, **TIFA**, and **LeX-Art**), together with confidence calibration to control the respective biases and hallucination risks of OCR and VLM.
>
> Specifically, it will include:
>
> * **Data construction:** a three-stage progressive design:
>
>   * **Short text / phrases:**
>   * **Multi-text / multi-region:**
>   * **Long text / dense text:**
>
> * **Annotated attributes:** the goal is to structure multiple pieces of target **visual text** into aligned, matchable, and localizable **GT**. Each sample will no longer correspond to a single holistic text string, but to a **label list**. Each **label** will contain at least **text**, **norm_policy**, **region**, **visibility**, and **style attributes** such as color, font, size, and orientation.
>
> * **Evaluation protocol:** we decompose text rendering into four measurable subproblems:
>
>   * **Visible text discovery:** whether text truly appears in the image;
>   * **Character content correctness:** whether the observed text matches the target prompt text;
>   * **Spatial and attribute satisfaction:** whether the text appears in the expected region and satisfies attributes such as color, font, and size;
>   * **“Generated text ↔ prompt label” correspondence:** in multi-text cases, which rendered text corresponds to which label.
>
> **4. Regarding CLT-Bench construction details and public release**
>
> Thank you for this helpful suggestion. The manuscript already gives the core principle of **CLT-Bench**: it is designed to stress-test Chinese text rendering under **rare-character** and **complex-layout** settings, with prompts stratified by a complexity score combining **character difficulty** and **layout difficulty**. We agree, however, that the current description is too brief. In the revision, we will expand the appendix with the full **CLT-Bench construction** pipeline, including data filtering, complexity stratification, and annotation examples. We also plan to publicly release the benchmark and its construction code to support reproducibility and future comparison.
>
> ---
>
> **We sincerely thank you again for your valuable time and constructive comments.**

---

> > ### Author Rebuttal · Reviewer_WXWT · 2026-04-02
> >
> > My concerns have been adequately addressed.

---

### Decision · Program_Chairs · 2026-04-30

**Decision:**

Accept (regular)

**Comment:**

The paper presents FreeText, a training free framework for improving text rendering in pre-trained text to image diffusion models without modifying model weights or imposing external layout constraints. The core idea is to decompose the problem into where to write and what to write, using cross attention based localization with sink like tokens to identify candidate text regions and Spectral Modulated Glyph Injection to strengthen glyph structure in the frequency domain. This design is positioned as a plug-n-play method that can be applied at inference time across multiple diffusion transformer architectures. The empirical study covers several base models, including FLUX, Qwen-Image, and SD-3, and reports consistent gains on readability oriented metrics across LongText Benchmark, CVTG, and others. The paper also contributes a benchmark (CLT-Bench) aimed at challenging Chinese long tail text rendering with more complex layouts and character distributions. Overall, the submission addresses an important and practically relevant weakness of current image generation systems with a method that is interesting, broadly applicable, and supported by experimental results.

The strengths of the paper are its clear problem formulation, well motivated decomposition of localization and glyph enhancement, training free deployment setting, and validation across several models while largely preserving semantic alignment and aesthetic quality. The main concerns raised by reviewers are also important, especially the lack of direct comparison to prior baselines, the under specification of CLT-Bench and some implementation details, and the need for clearer discussion of evaluation choices and limitations. At the same time, I find the rebuttal on the baseline issue reasonably convincing in context: the method is meant as an inference time enhancement for strong large scale foundation models, whereas several cited alternatives are retrained or layout dependent systems that operate differently and are already reported in prior work to lag behind the base models considered here. While the paper would be stronger with more explicit comparisons and fuller benchmark documentation, I believe the current contribution is sufficiently novel and useful to merit acceptance. My recommendation is to accept.